# Temperature-dependent modulation of light-induced circadian responses in *Drosophila melanogaster*

Yue Tian (ID) [1,2,3], Hailiang Li (ID) [1,2,3], Wenjing Ye[1,2,3], Xin Yuan[1,2,3], Xuan Guo (ID) [4] & Fang Guo (ID) [1,2,3 ✉]

## Abstract

**Animals entrain their circadian rhythms to multiple external signals, such as light and temperature, which are integrated in master clock neurons to adjust circadian phases. However, the precise mechanisms underlying this process remain unclear. Here, we use in vivo two-photon calcium imaging while precisely controlling temperature to investigate how the *Drosophila melanogaster* circadian clock integrates light and temperature inputs in circadian neurons. We show that light responses modulate the circadian clock in central pacemaker neurons, with temperature acting as a fine-tuning mechanism to achieve optimal adaptation. Our results suggest that temperature-sensitive dorsal clock neurons DN1as regulate the light-induced firing of s-LNv circadian pacemaker neurons and release of the neuropeptide PDF through inhibitory glutamatergic signaling. Specifically, higher temperatures suppress s-LNv firing upon light exposure, while lower temperatures enhance this response. Behavioral analyses further indicate that lower temperatures accelerate phase adjustment, whereas higher temperatures decelerate them in response to new light–dark cycles. This novel mechanism of temperature-dependent modulation of circadian phase adjustment provides new insights into the adaptive strategies of animals for survival in fluctuating environments.**

**Keywords** *Drosophila*; Circadian Rhythms; Photoentrainment; Temperature; Neural Circuit
**Subject Category** Neuroscience

## Introduction

The circadian rhythm system is an internal biological clock that enables organisms to adapt to the diurnal changes in their external environment. Regulatory mechanisms of circadian rhythms are remarkably conserved across species (Young and Kay, 2001). For instance, while the mammalian suprachiasmatic nucleus (SCN) contains tens of thousands of neurons (Welsh et al, 2010), the *Drosophila* brain comprises only 150 central circadian neurons(Herzog, 2007). Despite this simplicity, these neurons precisely regulate the sleep-wake behavioral cycle (Giebultowicz, 2018). Combined with the availability of abundant genetic tools (Lai and Lee, 2006; Ma and Ptashne, 1987; Nern et al, 2015; Nogi and Fukasawa, 1984; Potter et al, 2010), *Drosophila* has emerged as a pioneer model for investigating the neural circuits underlying circadian rhythms (Dubowy and Sehgal, 2017; Young and Kay, 2001).

The formation and maintenance of circadian rhythms rely on the autonomous regulation of the transcription-translation feedback loop (TTFL), an endogenous mechanism that generates rhythmic oscillations of core clock proteins (Patke et al, 2020). Beyond this intrinsic process, external environmental factors such as light and temperature also shape circadian rhythms (Buhr et al, 2010; Roenneberg and Foster, 1997). Among these, light is the most potent zeitgeber, regulating and resetting the circadian phase (Lee et al, 1996; Myers et al, 1996; Zeng et al, 1996). Light entrainment enables organisms to align their physiological and behavioral cycles with environmental changes, ensuring optimal adaptability. While extensive research has focused on light entrainment and synchronization (Chang, 2006; Evans, 2016; Nitabach and Taghert, 2008; Yoshii et al, 2016; Yoshii et al, 2015), the direct effects of light on central clock neurons and the neural circuits mediating light integration and transmission remain poorly understood. A recent study reported light-induced electrical responses in circadian neurons (Li et al, 2018), yet its use of in vitro methods limits the physiological relevance of the findings.

Temperature, another critical environmental zeitgeber, often collaborates with light to regulate circadian behavior in natural settings (Chen et al, 2018). However, most studies have focused on temperature's independent effects on circadian rhythms (Alpert et al, 2020; Alpert et al, 2022; Jin et al, 2021; Li et al, 2024; Miyasako et al, 2007), leaving the mechanisms by which light and temperature jointly influence circadian regulation largely unexplored. In natural environments, organisms constantly experience fluctuating light and temperature conditions. Understanding how these cues interact to shape circadian outputs is essential for elucidating the adaptive strategies that ensure survival in diverse environments.

In this study, we employed in vivo two-photon calcium imaging to investigate the light responses of central circadian neurons in *Drosophila*. Our findings revealed that s-LNvs, rather than other

[1]Department of Neurobiology, Department of Neurology of Sir Run Run Shaw Hospital and School of Brain Science and Brain Medicine, Zhejiang University School of Medicine, 310058 Hangzhou, China. [2]MOE Frontier Science Center for Brain Research and Brain-Machine Integration, State Key Laboratory of Brain-Machine Intelligence, Zhejiang University, 1369 West Wenyi Road, 311121 Hangzhou, China. [3]NHC and CAMS Key Laboratory of Medical Neurobiology, Zhejiang University, 310058 Hangzhou, China. [4]Life Science Institute, Jinzhou Medical University, 121001 Jinzhou, China. ✉E-mail: gfang@zju.edu.cn

circadian neurons, exhibit rhythmic responses to light stimulation under physiological conditions. These responses are modulated by the endogenous circadian clock, with smaller responses observed during the daytime and larger responses at night. Furthermore, we discovered that temperature significantly influences the firing patterns of s-LNvs in response to light: higher temperatures suppress light responses, while lower temperatures enhance them. This modulation is mediated by temperature-sensitive DN1a neurons, which transmit inhibitory glutamatergic signals to regulate s-LNv light responses. Crucially, our behavioral data demonstrated that temperature fine-tunes photoentrainment, with lower temperatures (19 °C) accelerating and higher temperatures (29 °C) slowing phase shifts to new light cycles. Notably, this process operates independently of Cryptochrome (CRY) in live flies. This temperature-dependent firing paradigm plays a vital role in modulating *Drosophila*'s light entrainment, shedding light on the intricate interplay between light and temperature in circadian phase adjustment.

# Results

## Investigation of circadian neuron light responses in live flies

To investigate the natural light responses of distinct circadian neuron subsets in live *Drosophila* under physiological conditions, we employed in vivo two-photon calcium imaging. This technique allowed us to monitor light pulse-induced activity changes in circadian neurons expressing GCaMP6s driven by the *CLK856-GAL4* across different Zeitgeber Time (ZT) points (Fig. 1A,B). The flies were entrained through multiple light–dark (LD) cycles before experimentation. To facilitate two-photon recordings at ZT6 and ZT18, we performed a brief surgical procedure to expose the dorsal brain by creating a small opening in the cuticle. This allowed us to observe the cell bodies of most dorsal circadian neurons, including DN1as, DN1ps, and DN2s, as well as the characteristic dorsal projection fibers of s-LNvs and LNds. For flies tested during the dark phase (ZT18), the surgery was conducted under dim red light to minimize disruption to the circadian clock during light-sensitive phases.

We first observed synchronized daily basal circadian calcium oscillations specifically in the dorsal fiber bundles of circadian neurons, rather than in their somata. These dorsal fiber bundles exhibited peak calcium levels at ZT6 and a trough at ZT18 (Fig. EV1A), contrasting with the higher calcium levels observed in the soma at ZT18 (Fig. 1C). This suggests that the calcium level in circadian neuron cell bodies and fibers may be subject to distinct circadian regulatory mechanisms.

Interestingly, the circadian neurons displayed minimal light-induced calcium activity at ZT6, indicating a temporal "dead zone" during the daytime (Fig. 1C). Among all circadian neurons examined, the morning cells (M cells), particularly the s-LNvs, exhibited the most robust response to light stimulation at ZT18 (Fig. 1D). This response was marked by an almost one-fold increase in GCaMP signal during light exposure, followed by a significant reduction in calcium activity of the s-LNvs to levels below baseline (Fig. 1E). These findings suggest that s-LNvs function as primary

recipients of visual input within the circadian circuit, with their light responses gated by the circadian clock.

## Circadian dynamics of light responses in s-LNvs

Since the data from *CLK856-GAL4 > GCaMP* flies does not distinctly separate the fibers of s-LNvs from other circadian neurons (such as DN1a and ITP+ LNd) that also project to the accessory medulla (aMe), we further characterized the light responses of s-LNvs using *Dvpdf-LexA*, which also labels the dorsal projection fiber of s-LNvs (Fig. 2A), and the *SS00681* split-gal4, which specifically marks s-LNv neurons only. We assessed the light responses of s-LNvs in live flies at four circadian time points (ZT0-2, ZT5-7, ZT11-13, ZT17-19) under the two-photon microscope.

To comprehensively investigate whether s-LNvs exhibit circadian spectral sensitivity to light, we exposed flies to light pulses at three different wavelengths (blue, green, and yellow). Our recordings revealed significant circadian variations in the s-LNvs' responses to light stimulation. Specifically, s-LNvs fibers exhibited higher basal calcium levels and minimal light-induced activity during the day, whereas they showed lower basal calcium levels and more pronounced light-induced activation at night (Figs. 2B–D and EV1B–D; Movies EV1 and EV2). While ΔF/F shows a clear diurnal rhythm, our $F_{peak}$ data does not show a similar circadian oscillation (Fig. EV1E). Therefore, the observed rhythm in ΔF/F is substantially influenced, and potentially primarily driven by the diurnal fluctuations in baseline calcium levels. Notably, we did not observe significant wavelength-dependent differences in s-LNvs responses at any time point (Fig. 2E; Table EV1). Interestingly, s-LNvs consistently exhibited significantly reduced calcium signals following light-induced activation, dropping below baseline levels (Fig. 2E). This distinct calcium dynamics may underlie the bursting properties of s-LNvs (Fourcaud-Trocmé et al, 2022), as such firing in bursts has been associated with PDF neuropeptide release (Fernandez-Chiappe et al, 2021).

## Photoreceptor pathways mediating circadian light responses in s-LNvs

We next investigated the photoreceptors that provide visual inputs to s-LNvs in *Drosophila*. *Drosophila* possesses multiple photoreceptors, including external ones such as compound eyes, ocelli, and Hofbauer–Buchner (H–B) eyelets, as well as the circadian photoreceptor CRY (Fig. 3A). The *norpA* gene, which encodes phosphatidylinositol-specific phospholipase C, is crucial for phototransduction in the eyes. Mutation of *norpA* renders *Drosophila* effectively visual-blind (Zhu et al, 1993). Among these photoreceptors, H–B eyelets are particularly notable as they directly project to s-LNvs and surrounding areas (Yoshii et al, 2016). To separately determine the contribution of these photoreceptors, we measured light responses of s-LNv neurons in both *norpA* mutant flies and flies with ablated H–B eyelets. In *norpA* mutants, the s-LNv response to light stimulation was almost completely abolished at all ZT time points (Fig. 3B; Table EV2). In flies with ablated H–B eyelets, the amplitude of the s-LNv light response was significantly reduced, specifically during nighttime (Fig. 3C; Table EV2). These results indicate that the light response of s-LNv neurons primarily originates from visual input pathways.

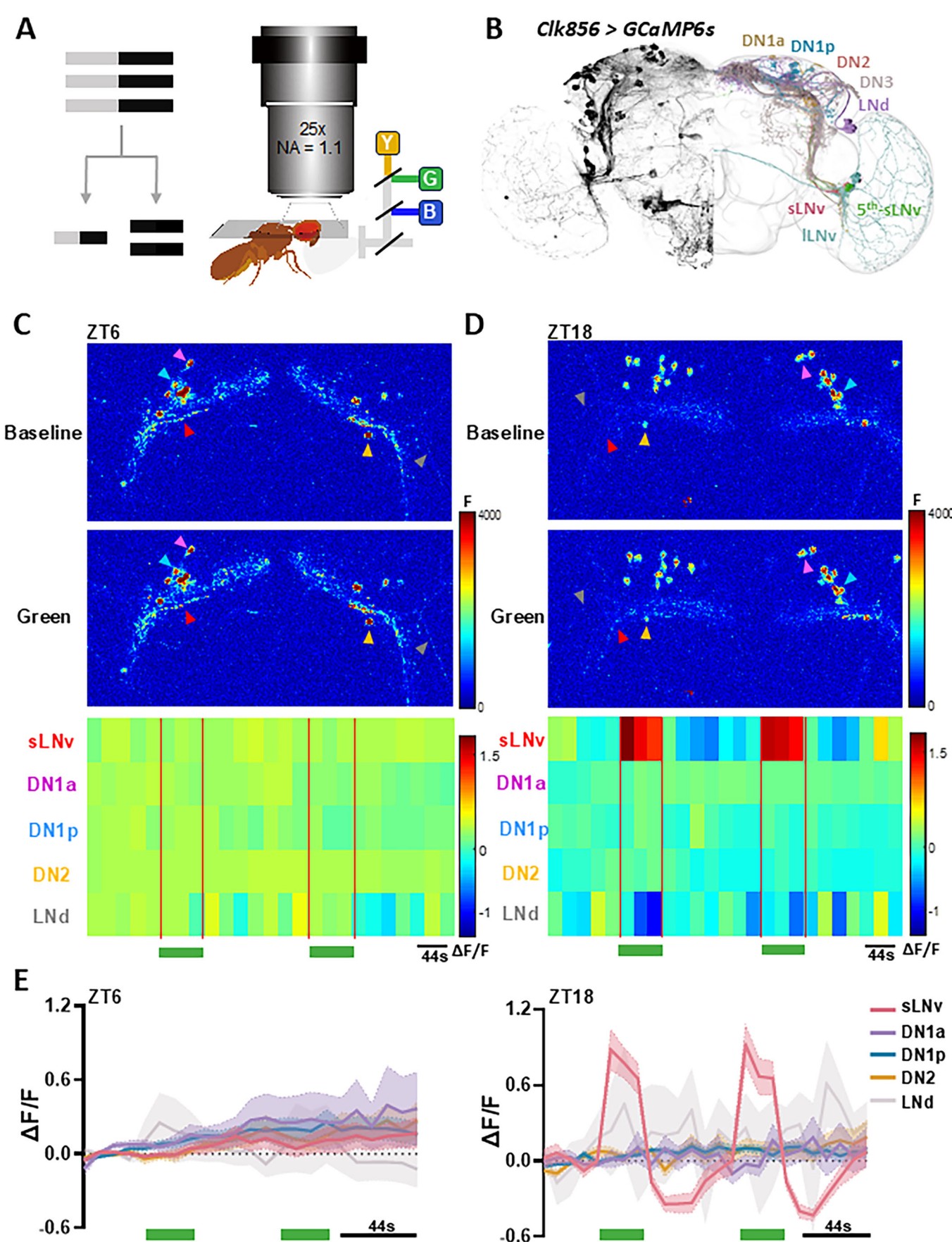

**Figure 1. Circadian neuron light responses in live flies.**

(A) Schematic diagram of a two-photon calcium imaging device. (B) Schematic diagram of circadian neurons. The left is an immunohistochemistry diagram of circadian neurons expressing GCaMP labeled by *CLK856-GAL4*; the right is an electron microscope data display obtained from FlyWire dataset. (C, D) Up: pseudocolor map of the calcium activity changes of circadian neurons during the day ZT6 (C) and at night ZT18 (D) in response to light stimulation. Bottom: $\Delta F/F_O$ heatmap of the response of circadian neurons to light stimulation. (E) Traces of $\Delta F/F_O$ of the s-LNvs dorsal fibers at different time points (Left: ZT6, Right: ZT18) in response to light pulse, expressed as mean ± standard error. $N = 4$ for each ZT. Source data are available online for this figure.

In *Drosophila*, environmental light especially the blue light is also detected by the CRY, which is expressed in key clock neurons, including s-LNvs. Previous studies have demonstrated that *cry^b* mutants exhibit impaired photoentrainment to LD cycles (Emery et al, 2000; Helfrich-Forster et al, 2001; Stanewsky et al, 1998). Furthermore, CRY functions as a blue-light sensor, increasing neuronal firing in light-responsive circadian neurons, and its expression levels fluctuate in key circadian neurons in a time-dependent manner (Kistenpfennig et al, 2012). To investigate whether CRY regulates the rhythmicity of s-LNvs' light response, we recorded light responses in *cry* mutant flies. Surprisingly, neither the immediate light response nor its rhythmicity in s-LNvs was affected in *cry* mutants compared to wild-type flies (Fig. 3D, Table EV2). These findings suggest that CRY is not essential for the circadian response of s-LNvs to short, acute light pulses.

## Endogenous circadian regulation of light responses in s-LNvs

We then examined whether l-LNvs are an alternative light input pathway to s-LNvs. l-LNvs are light-responsive neurons communicating with s-LNvs through Pigment Dispersing Factor (PDF) and PDF receptor (PDFR) signaling (Shafer et al, 2008; Shang et al, 2008). Given the observed circadian differences in s-LNv light response, we examined whether upstream l-LNvs also exhibit similar circadian patterns. Using an l-LNv-specific split-GAL4 line, we recorded l-LNvs calcium responses to light stimulation. Notably, l-LNvs showed consistent moderate light-induced activation across all four ZT time points (Figs. 4A and EV2). Together, our data demonstrated that the day–night variation in s-LNv light response originates within the circadian neurons themselves rather than through upstream visual pathways.

Since s-LNv neurons express PDFR and receive PDF input from l-LNvs, as well as the PDFR expression in s-LNvs may undergo circadian regulation (Ma et al, 2021), we next examined whether PDFR regulates the rhythmicity of the s-LNvs light response. A comparison of light response characteristics between PDFR mutants and wild-type controls revealed that the rhythmicity of the s-LNvs light response remained intact (Fig. 4B; Table EV3).

We hypothesized that the circadian rhythmicity in s-LNvs light responses might arise from the endogenous circadian clock modulating basal calcium levels across day and night. This predicts that higher basal calcium levels during daytime could limit light-induced activation through a ceiling effect, while lower nighttime levels could permit larger light-induced calcium increases. To test this hypothesis, we measured basal calcium levels in s-LNvs *per* mutant flies, which lack a functional circadian clock. As predicted, *per^0* mutants showed no day–night variation in basal calcium levels, maintaining consistently low GCaMP signals in s-LNv fibers under LD (Fig. EV3).

To definitively establish whether the endogenous circadian clock mediates s-LNvs light response rhythmicity, we recorded light responses in *per^0* mutants under both LD and DD conditions. The rhythmicity of the s-LNvs light response was abolished in *per^0* mutants under both conditions. Under LD conditions, s-LNvs in *per^0* mutants exhibited consistent, moderate responses to light stimulation across all time points (Fig. 4C,D; Table EV3). However, under DD conditions, these neurons showed virtually no response to light stimulation, regardless of circadian time (Fig. 4E,F; Table EV3). In contrast, wild-type controls maintained rhythmic s-LNvs light responses in DD, demonstrating that the endogenous circadian clock, rather than environmental light cycles, sustains this rhythmicity. Collectively, these results establish that the endogenous circadian clock is essential for maintaining circadian light responses in s-LNvs.

## Neuronal circuit mechanisms underlying temperature-induced modulation of light responses in s-LNvs

In natural environments, ambient light and temperature function as key coupled zeitgebers. Although environmental temperature cycles typically lag behind light cycles by several hours (Vanin et al, 2012), it remains unclear whether these two signals interact synergistically or antagonistically on key circadian neurons. To address this question, we investigated how s-LNvs calibrate their light-induced firing and PDF peptide release in response to environmental temperature variations. We employed our newly developed temperature control system in combination with two-photon in vivo calcium imaging to record the response patterns of s-LNvs dorsal fibers to light pulses at different temperatures and circadian time points (Li et al, 2024).

The experimental protocol comprised both cooling and heating procedures. In the cooling protocol, we measured light pulse responses in the same flies sequentially at 24 °C, 15 °C, and again at 24 °C. For the heating protocol, we tested responses at 22 °C, 31 °C, and again at 22 °C. Interestingly, flies exhibited enhanced s-LNvs light responses at low temperatures, with increased calcium levels during light pulses at both noon (ZT6) and middle night (ZT18) (Fig. 5A–C). Conversely, higher temperatures inhibited s-LNvs light responses, with significantly reduced responses at ZT18 (Fig. 5D–F). Nonetheless, two possibilities could explain these findings: (1) genuine changes in light responsiveness or (2) artifacts arising from fluctuating baseline calcium levels, which the $\Delta F/F$ data presentation does not allow us to differentiate. To address this critical issue, we analyzed the basal calcium fluorescence values ($F_{baseline}$) in s-LNv fibers before light stimulation and the maximum fluorescence values ($F_{peak}$) during light stimulation across temperature changes at different circadian time points (ZT6, ZT18).The results showed that for each fly tested at a specific time point, there was no significant difference in the basal fluorescence

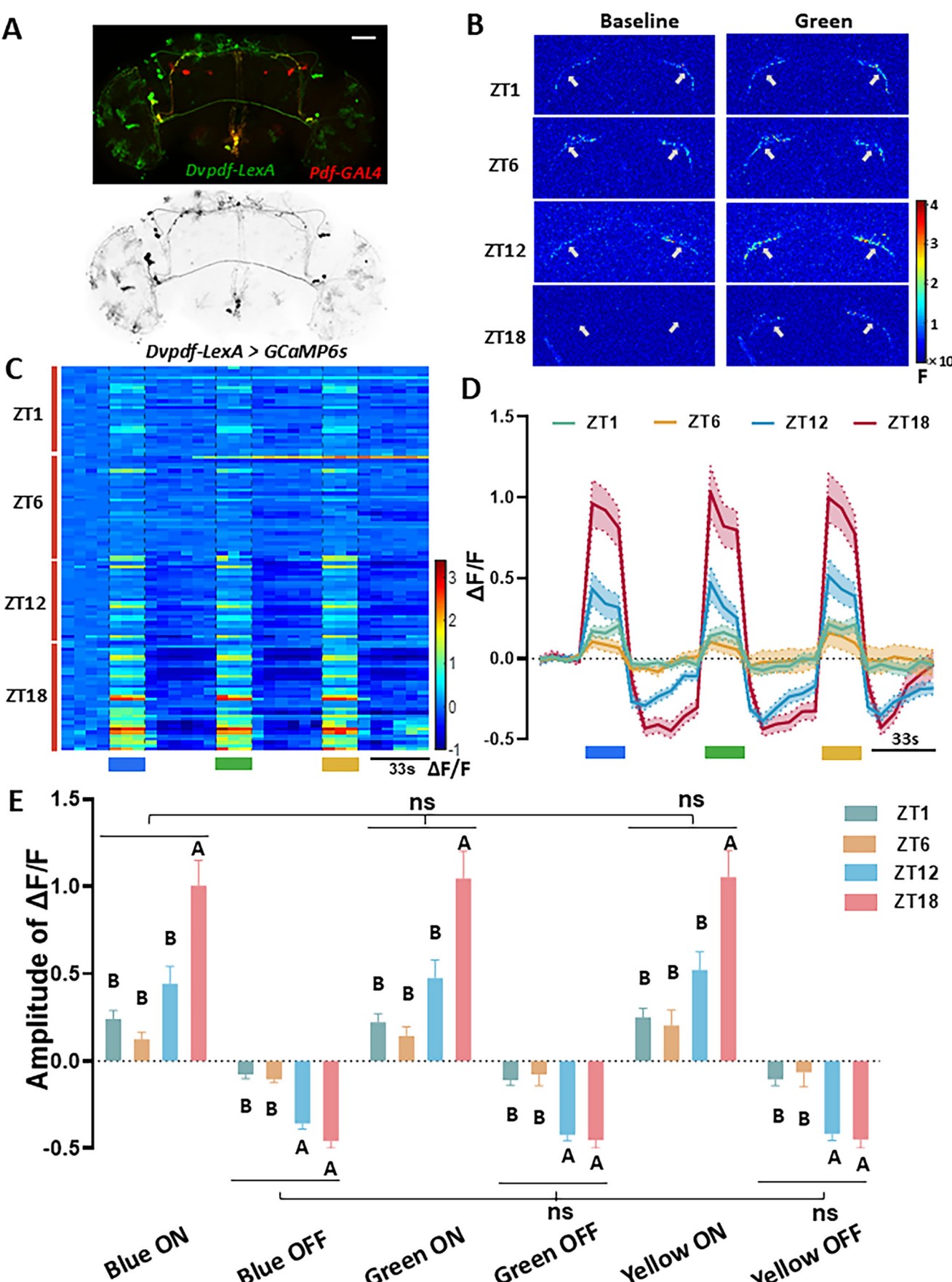

**Figure 2. Circadian dynamics of light responses in s-LNvs.**

(A) Representative double-staining images of *Dvpdf-LexA* expressing GCaMP and *PDF-GAL4* expressing tdTomato in fly brains. Brains were stained with anti-GFP (green) and anti-tdTomato (red). Scale bar = 50 μm. (B) Pseudocolor map of the calcium activity changes of the s-LNvs dorsal fibers at different time points in response to a light pulse. (C) $\Delta F/F_O$ heatmap of the light response of s-LNvs dorsal fibers at different time points. Each row represents an ROI. Generally, each brain has two ROIs, one for each of the left and right hemispheres. (D) Traces of $\Delta F/F_O$ s-LNvs at different time points in response to a light pulse. (E) Statistical quantification of the response amplitudes of s-LNvs at different time points to different colors. $N = 13$ for ZT1, $N = 16$ for ZT6, $N = 14$ for ZT12, and $N = 17$ for ZT18. Data information: Data are presented as means ± SEM. Statistical significance was compared within different ZTs and different lights. A two-way ANOVA of blue, green, and yellow ON, followed by another ANOVA with OFF, was used. The letters A and B above or below the histograms denote significantly different means within each of the two groups, $P < 0.05$. Source data are available online for this figure.

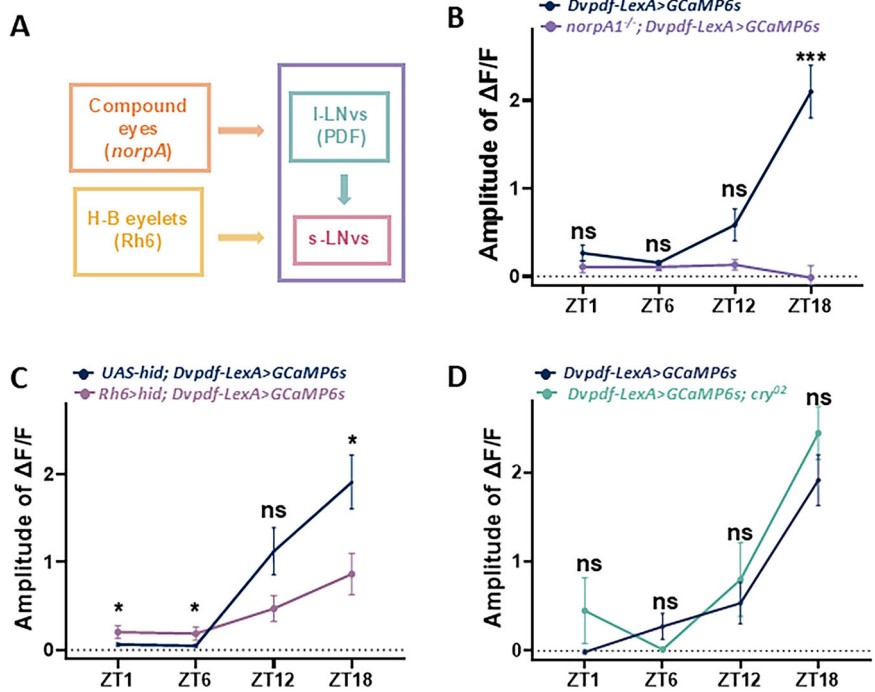

**Figure 3. Photoreceptor pathways mediating light responses in s-LNvs.**

(A) Schematic illustration of visual inputs to the pacemaker neurons. (B) The calcium response amplitude of the s-LNvs dorsal fibers in wild-type control flies (dark blue) and *norpA*$^{-/-}$ mutant (purple) at different time points in response to a light pulse. The number of wild-type control flies was $N = 4$ for ZT1 and ZT6, $N = 5$ for ZT12, and $N = 6$ for ZT18. The number of *norpA*$^{-/-}$ mutant was $N = 4$ for ZT1, ZT6, and ZT12, and $N = 3$ for ZT18. The Benjamini–Hochberg FDR-adjusted $P$ value was $P = 0.0008$ for ZT18. (C) The calcium response amplitude of the s-LNvs in wild-type control flies (dark blue) and flies with apoptotic H–B eyelets (pink) at different time points in response to a light pulse. The number of wild-type control flies was $N = 8$ for ZT1 and ZT12, and $N = 7$ for ZT12 and ZT18. The number of H–B eyelets ablated flies was $N = 4$ for each ZT. The Benjamini–Hochberg FDR-adjusted $P$ value was $P = 0.0411$ for ZT1, ZT6, and ZT18. (D) The calcium response amplitude of the s-LNvs in wild-type control flies (dark blue) and *cry*$^{02}$ mutant (green) at different time points in response to light stimulation. The number of wild-type control flies was $N = 3$ for ZT1, $N = 4$ for ZT6, ZT12, and ZT18. The number of *cry*$^{02}$ mutants was $N = 3$ for ZT1, ZT12, and $N = 4$ for ZT6 and ZT18. Data information: Data are presented as means ± SEM. *$P < 0.05$, ***$P < 0.001$. Statistical significance was compared between wild-type control flies and experimental flies at each ZT using a two-tailed unpaired Student's $t$ test with post hoc Benjamini–Hochberg FDR correction. One-way ANOVA was used in wild-type control flies and experimental flies separately for four different time points. Source data are available online for this figure.

values before, during, and after temperature changes (Fig. EV4A–D). Importantly, $F_{peak}$ during light stimulation was significantly higher at lower temperatures compared to normal temperatures (Fig. EV4E,F), while at higher temperatures, $F_{max}$ was significantly lower than at normal temperatures (Fig. EV4G,H). These findings indicate that the light response amplitude (as measured by $F_{max}$) changes with temperature throughout the day. They further suggest that key circadian pacemakers employ a compensatory mechanism to modulate their light response based on temperature—reducing light-induced pacemaker firing during hot seasons and enhancing responses during colder seasons.

We next investigated the underlying neuronal circuit mechanism of this compensatory effect. Recent findings have identified dorsal neurons 1a (DN1as) as temperature-sensitive circadian neurons within the circadian network. DN1as respond to both cooling and heating in a circadian-dependent manner: they are significantly inhibited by low temperatures and excited by high temperatures (Alpert et al, 2020; Li et al, 2024). The FlyWire electron microscopy connectomic database of the *Drosophila* brain reveals possible peptidergic connections between temperature-sensitive DN1as and light-sensitive s-LNvs (Fig. 6A). Furthermore, s-LNvs and DN1as axons exhibit reciprocal compensatory circadian rewiring patterns,

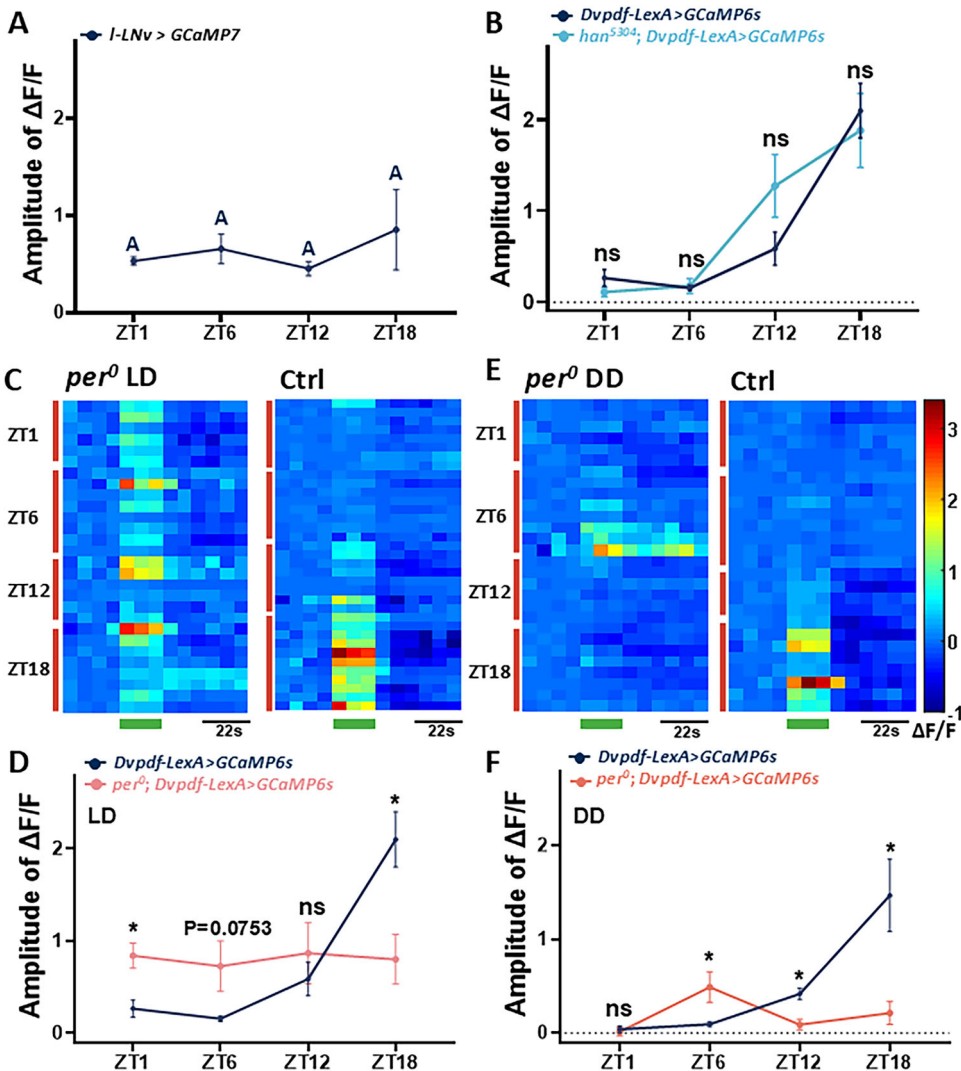

**Figure 4. Endogenous circadian pathways mediating light responses in s-LNvs.**

(A) The calcium response amplitude of l-LNvs in wild-type control flies. $N = 4$ for ZT1, ZT6, and ZT12, and $N = 3$ for ZT18. (B) The calcium response amplitude of s-LNvs in wild-type flies (dark blue) and *PDFR* mutant flies (blue) at different time points in response to a light pulse. The number of wild-type control flies was $N = 4$ for ZT1 and ZT6, $N = 5$ for ZT12, and $N = 6$ for ZT18. The number of *PDFR* mutants was $N = 4$ for ZT1, ZT6, and ZT12, and $N = 5$ for ZT18. (C) Heatmap of the calcium activity of s-LNvs in response to a light pulse at different circadian time points in *per^0* mutant (left) and wild-type control *Drosophila* (right) under LD conditions. (D) The calcium response amplitude of the s-LNvs in wild-type control flies (dark blue) and *per^0* mutant (red) under LD conditions. The number of wild-type control flies was $N = 4$ for ZT1 and ZT6, $N = 5$ for ZT12, $N = 6$ for ZT18. The number of *per^0* mutant was $N = 4$ for ZT1, ZT6, and ZT18, $N = 3$ for ZT12. $P = 0.0034$ at ZT1, $P = 0.0066$ at ZT18. The Benjamini–Hochberg FDR-adjusted $P$ value was $P = 0.0132$ for ZT1 and ZT18. (E) Heatmap of the calcium activity of s-LNvs in response to a light pulse at different circadian time points in *per^0* mutant (left) and wild-type control *Drosophila* (right) under DD conditions. (F) The calcium response amplitude of the s-LNvs in wild-type control flies (dark blue) and *per^0* mutant (red) under DD conditions. The number of wild-type control flies was $N = 4$ for ZT6 and ZT18, $N = 3$ for ZT1 and ZT12. The number of *per^0* mutants was $N = 3$ for ZT1 and ZT12, $N = 4$ for ZT6 and ZT18. $P = 0.0078$ at ZT18. The Benjamini–Hochberg FDR-adjusted $P$ value was $P = 0.0428$ for ZT6 and $P = 0.0156$ for ZT12 and ZT18. Data information: Data are presented as means ± SEM. Statistical significance was compared using One-way ANOVA in (A, B, D, F). Two-tailed unpaired Student's *t* test with post hoc Benjamini–Hochberg FDR correction between wild-type control flies and experimental flies was used in (B, D, F) at each ZT. Source data are available online for this figure.

where the maximum presynaptic area in one population coincides with the minimum in the other (Song et al, 2021). These findings led us to hypothesize that temperature may act through DN1as to negatively modulate s-LNvs light responses.

To test this hypothesis, we first verified whether inhibitory functional connections exist from DN1as to s-LNvs using optogenetically coupled two-photon in vivo calcium imaging. To avoid confounding effects from optogenetic red light pulses on s-LNvs, we conducted this experiment in a *norpA1* genetic background. When DN1as expressing the light-controlled cation channel CsChrimson were activated by 630 nm red light, the calcium activity of s-LNvs decreased significantly (Fig. 6B,C), confirming that DN1as exert an inhibitory effect on s-LNvs.

We then investigated whether DN1a neuronal activity affects s-LNv light responses. We examined s-LNvs responses to light stimulation in flies where DN1as were ablated through the specific expression of the

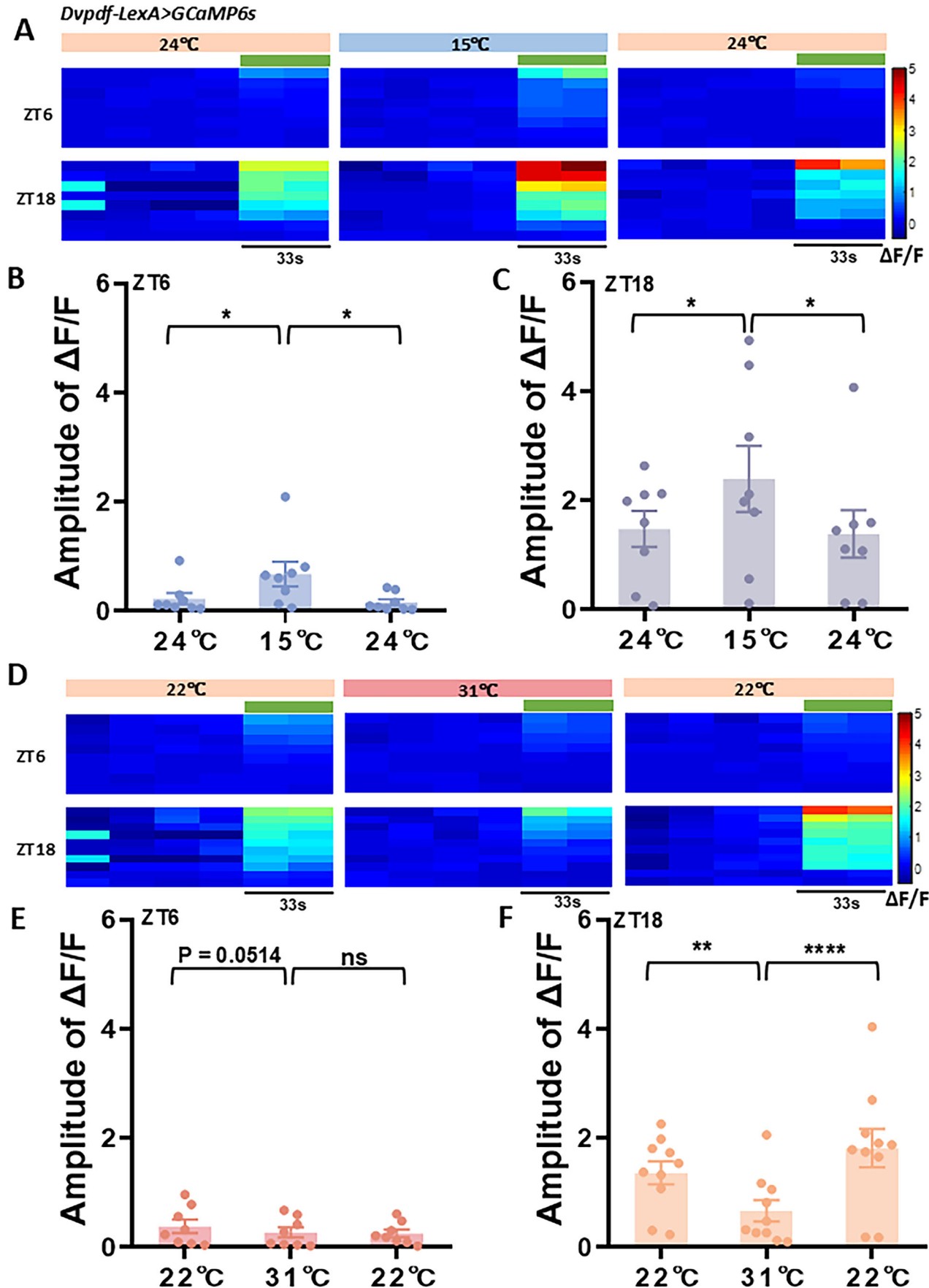

◄ Figure 5.   Temperature-induced modulation of light responses in s-LNvs.

(A) ΔF/F_O heatmap of the calcium activity of s-LNvs at room temperature (24 °C) and low temperature (15 °C) at ZT6 and ZT18 in response to a light pulse.
(B, C) Statistical quantification of the calcium response amplitude of s-LNvs at 24 °C and 15 °C at ZT6 (B) and ZT18 (C) in response to a light pulse. $N = 6$ for ZT6, $N = 4$ for ZT18. $P = 0.0316$ for 24 °C vs 15 °C, $P = 0.0155$ for 15 °C vs 24 °C in (B). $P = 0.0398$ for 24 °C vs 15 °C, $P = 0.0259$ for 15 °C vs 24 °C in (C). (D) ΔF/F_O heatmap of the calcium of s-LNvs at normal temperature (22 °C) and high temperature (31 °C) at ZT6 and ZT18 in response to a light pulse. (E, F) Statistical quantification of the calcium response amplitude of s-LNvs at 22 °C and 31 °C at ZT6 (E) and ZT18 (F) in response to a light pulse. $N = 4$ for ZT6, $N = 7$ for ZT18. $P = 0.0044$ for 22 °C vs 31 °C, $P < 0.0001$ for 31 °C vs 22 °C in (F). Data information: Data are presented as means ± SEM. *$P < 0.05$, **$P < 0.01$, ****$P < 0.0001$. Statistical significance was compared with 15 °C (B, C) and 31 °C (E, F). One-way ANOVA was used. Source data are available online for this figure.

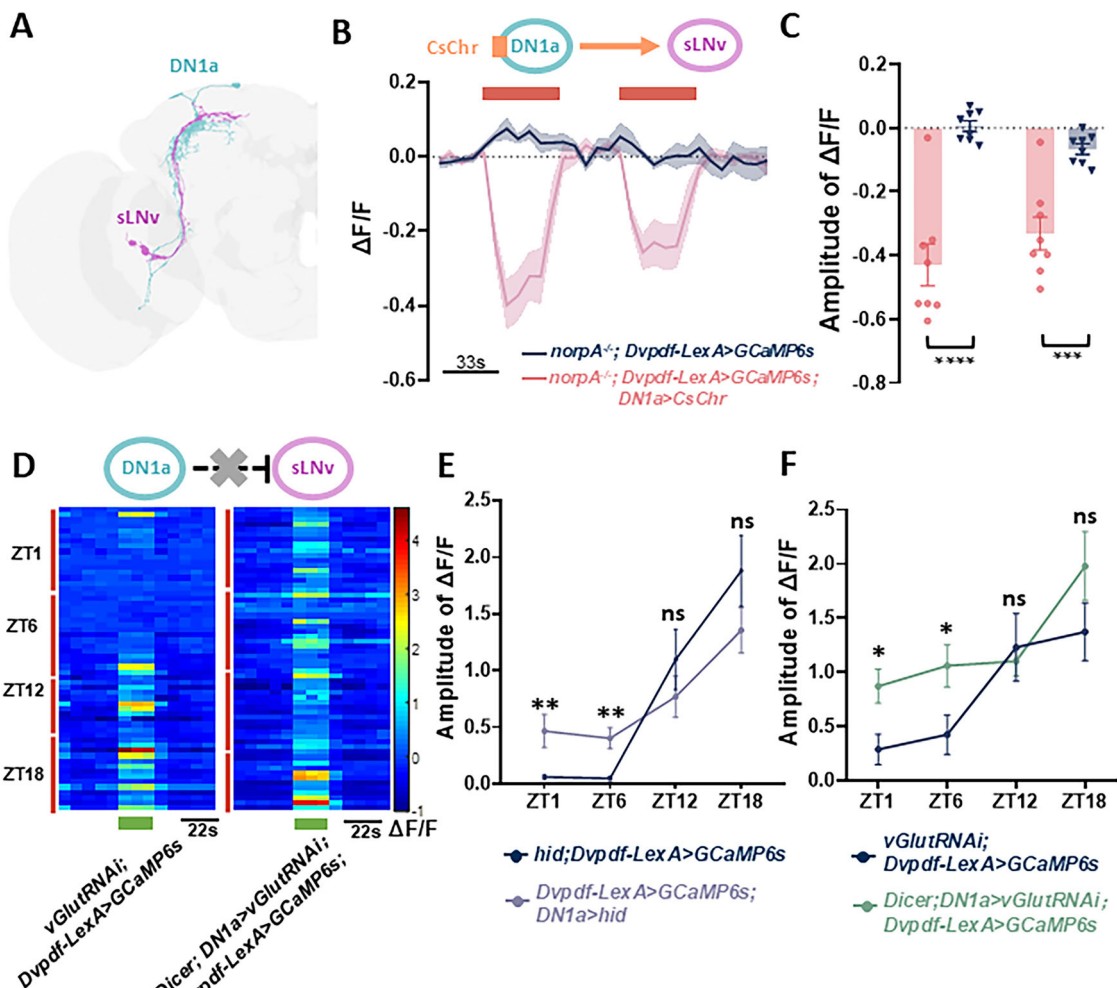

Figure 6.   DN1as exert glutamatergic inhibition to s-LNvs.

(A) The electron microscopy connection diagram of DN1as and s-LNvs; the cyan neurons are DN1as; the pink neurons are s-LNvs. (B) Up: schematic diagram of optogenetics. The photosensitive protein Cschrimson is expressed in the upstream neurons DN1as, and the calcium indicator GCaMP6s is expressed in the downstream neurons s-LNvs. Bottom: representative ΔF/F_O traces of the calcium activity of s-LNvs neurons in wild-type control flies (gray) and flies with DN1as activation by red light (red). $N = 4$ for both control flies and DN1as-activated flies. (C) Statistical quantification of the response amplitude of s-LNvs neurons in wild-type control flies (gray) and flies with DN1as activation by red light (red). $N = 4$ for both control flies and DN1as-activated flies. $P < 0.0001$ for the first stimulus, $P = 0.0002$ for the second stimulus. (D) Heatmap of the calcium activity of s-LNvs in response to a light pulse at different circadian time points in wild-type control flies(left) and DN1-inhibited flies (right). (E) The calcium response amplitude of the s-LNvs in wild-type control flies (dark blue) and flies with apoptotic DN1as (gray purple). The number of wild-type control flies was $N = 8$ for ZT1 and ZT12, $N = 7$ for ZT6 and ZT18. The number of DN1as ablated flies was $N = 6$ for ZT1, $N = 7$ for ZT6, $N = 5$ for ZT12, and $N = 9$ for ZT18. The Benjamini–Hochberg FDR-adjusted $P$ value was $P = 0.0074$ for ZT1 and $P = 0.0020$ for ZT6. (F) The calcium response amplitude of the s-LNvs in wild-type control flies (dark blue) and flies with RNAi-mediated knockdown of glutamate transmission in DN1as (green). The number of wild-type control flies was $N = 8$ for ZT1 and ZT6, $N = 6$ for ZT12, and $N = 10$ for ZT18. The number of DN1a-inhibited flies was $N = 8$ for ZT1, ZT6, and ZT12, and $N = 6$ for ZT18. The Benjamini–Hochberg FDR-adjusted $P$ value was $P = 0.0376$ for ZT1 and $P = 0.0460$ for ZT6. Data information: Data are presented as means ± SEM. Statistical significance was compared between wild-type control flies and experimental flies at each ZT using a two-tailed unpaired Student's $t$ test with post hoc Benjamini–Hochberg FDR correction. One-way ANOVA was used in wild-type control flies and experimental flies separately for four different time points. Source data are available online for this figure.

apoptotic protein HID. As expected, the elimination of DN1as, which mimics a low-temperature environment, enhanced s-LNvs light responses, particularly during the day (Fig. 6E; Table EV4). The previous study reported that the s-LNv dorsal termini puncta are directly inhibited by Glutamate(Fernandez et al, 2020). Given that DN1as are glutamatergic neurons, we also employed RNAi-mediated knockdown of glutamate transmission in DN1as and observed concordant results: DN1a-inhibited flies consistently exhibited increased light-induced s-LNv activation during daytime, with a slight but non-significant increase during nighttime periods (Fig. 6D,F; Table EV4). Meanwhile, we calculated both $F_{baseline}$ and $F_{peak}$ of s-LNv fibers in wild-type control flies and DN1a-inhibited flies. Consistent with our temperature findings, DN1a-inhibited flies exhibited an increased maximum GCaMP signal in response to light at most ZT time points (Fig. EV5A), while baseline values remained comparable to controls (Fig. EV5B). Collectively, these findings demonstrate that temperature may regulate s-LNvs light responses through DN1a-mediated glutamatergic inhibition.

## Temperature fine-tuned the light-induced phase shifts

Having established the neural circuit mechanisms by which temperature modulates s-LNvs light responses, we next sought to explore how these physiological changes influence circadian behaviors. Given the well-established role of s-LNvs in regulating circadian rhythms, we hypothesized that temperature-dependent changes in light response in s-LNv could significantly affect light entrainment. The s-LNvs express PDF, a neuropeptide that acts as a critical circadian regulator (Helfrich-Forster, 1995). Notably, artificial stimulation of s-LNvs firing during dark periods promotes PDF release, reduces circadian clock protein levels in downstream neurons, and induces phase shifts in the circadian clock under constant darkness (DD) via a CRY-independent pathway (Eck et al, 2016; Guo et al, 2014).

Despite extensive research on PDF's role in photo response and photoentrainment, most studies have been conducted under constant temperature conditions (Guo et al, 2014; Lamba et al, 2014; Schlichting et al, 2016; Vaze and Helfrich-Förster, 2021; Yoshii et al, 2009). This leaves a significant gap in understanding how temperature specifically modulates PDF's contribution to photoentrainment. This question is especially relevant in natural environments, where fruit flies experience daily and seasonal fluctuations in temperature. The interplay between temperature changes and light in regulating circadian rhythms and PDF function remains poorly understood.

Our findings reveal that ambient temperature modulates light-induced s-LNvs activity and PDF release, potentially leading to differential phase shifts. To isolate the influence of light-induced s-LNvs firing on phase shifts from CRY-mediated photoentrainment, we conducted phase shift experiments in both wild-type flies and *cry* mutants. Our results already showed that *cry* mutations do not affect light-induced s-LNvs activation at different zeitgeber time (ZT) points (Fig. 3D). To investigate temperature's impact on phase delays or advances, we tested flies at three temperatures: 24 °C (room temperature), 19 °C (low temperature), and 29 °C (high temperature). Wild-type and *cry* mutant flies were first synchronized to identical LD cycles to align their circadian phases. Subsequently, these flies were subjected to either an 8-hour phase advance or a delay of the LD cycle. This experimental design allowed us to systematically evaluate how temperature influences the circadian system's ability to adapt to phase shifts.

Average actograms for all strains are presented in Figs. 7A and EV6A. Following an 8-hour phase delay of the LD cycle at 24 °C, wild-type flies rapidly adjusted their activity rhythms, stabilizing their phase within 2 days (Fig. 7A). In contrast, *cry^02^* mutants required approximately 4 days to fully synchronize to the new LD cycle at 24 °C, consistent with previous findings (Yoshii et al, 2015). To quantify the rate of phase delays at different temperatures, we measured the slope of the evening activity peak under the new LD scheme. Notably, wild-type flies achieved near-complete re-entrainment by the second day of the new LD cycle across all tested temperatures (19, 24, and 29 °C) (Fig. 7B,C). However, on the first day of the new LD cycle, their phase shifts exhibited a clear temperature-dependent pattern. Compared to flies at 24 °C, those at 19 °C displayed phase delays 50 min faster, while those at 29 °C showed phase delays 50 min slower (Fig. EV6B,C). This suggests a potential role for temperature-modulated s-LNv firing, acting in parallel with CRY, in mediating phase shifts in wild-type flies.

Notably, *cry^02^* mutants exhibited 2 h slower phase delays at 29 °C than at 24 °C on day 1 to day 4 of the new LD scheme (Fig. 7B; Table EV5), suggesting that reduced light-induced s-LNvs firing amplitude at higher temperatures slows phase shifts in response to new LD cycles. At lower temperatures (19 °C), *cry^02^* mutants showed a phase delay comparable to that observed at 24 °C (Fig. 7C; Table EV6). These temperature-dependent effects suggest that thermal conditions influence phase delay dynamics in *cry^02^* mutants through modulation of s-LNvs activity.

We also examined responses to phase advances, finding that *cry^02^* mutants displayed slower phase advances at 29 °C but more rapid advances at 19 °C when subjected to an 8-hour phase advance of the LD cycle (Fig. 7D,E; Tables EV7 and EV8). Differences in the evening peak onset slope in the new LD scheme evidenced this temperature sensitivity (Fig. EV6A,D,E). Wild-type flies, in contrast, exhibited similar rates of phase advance across all temperatures. Collectively, these data support our model that temperature modulates light-induced phase shifts by calibrating s-LNvs light responses and potentially PDF release in a CRY-independent manner (Fig. 7F).

# Discussion

In this study, we investigated light pulse-induced activity changes in distinct circadian neuron subsets under near-physiological conditions in living flies. Our findings revealed that the M cell s-LNvs exhibit the most pronounced response to light stimulation, characterized by typical circadian dynamics. This light response originates from visual inputs and is regulated by the endogenous circadian clock, supporting the model that s-LNvs serve as primary recipients of visual input, with their light sensitivity finely tuned by the circadian clock.

Using a two-photon microscope coupled with a temperature control system, we uncovered a novel finding: the firing pattern of s-LNvs in response to light pulses is significantly influenced by external ambient temperature. We identified the neuronal circuit underlying this modulation, demonstrating that temperature-sensitive DN1a neurons convey inhibitory glutamatergic signals to regulate the light response of s-LNvs. Interestingly, DN1a ablation or DN1a>VGLUT RNAi enhanced light pulse responses only during the day but not at ZT18, in contrast to the pattern in wild-type flies under actual low temperatures. This discrepancy is

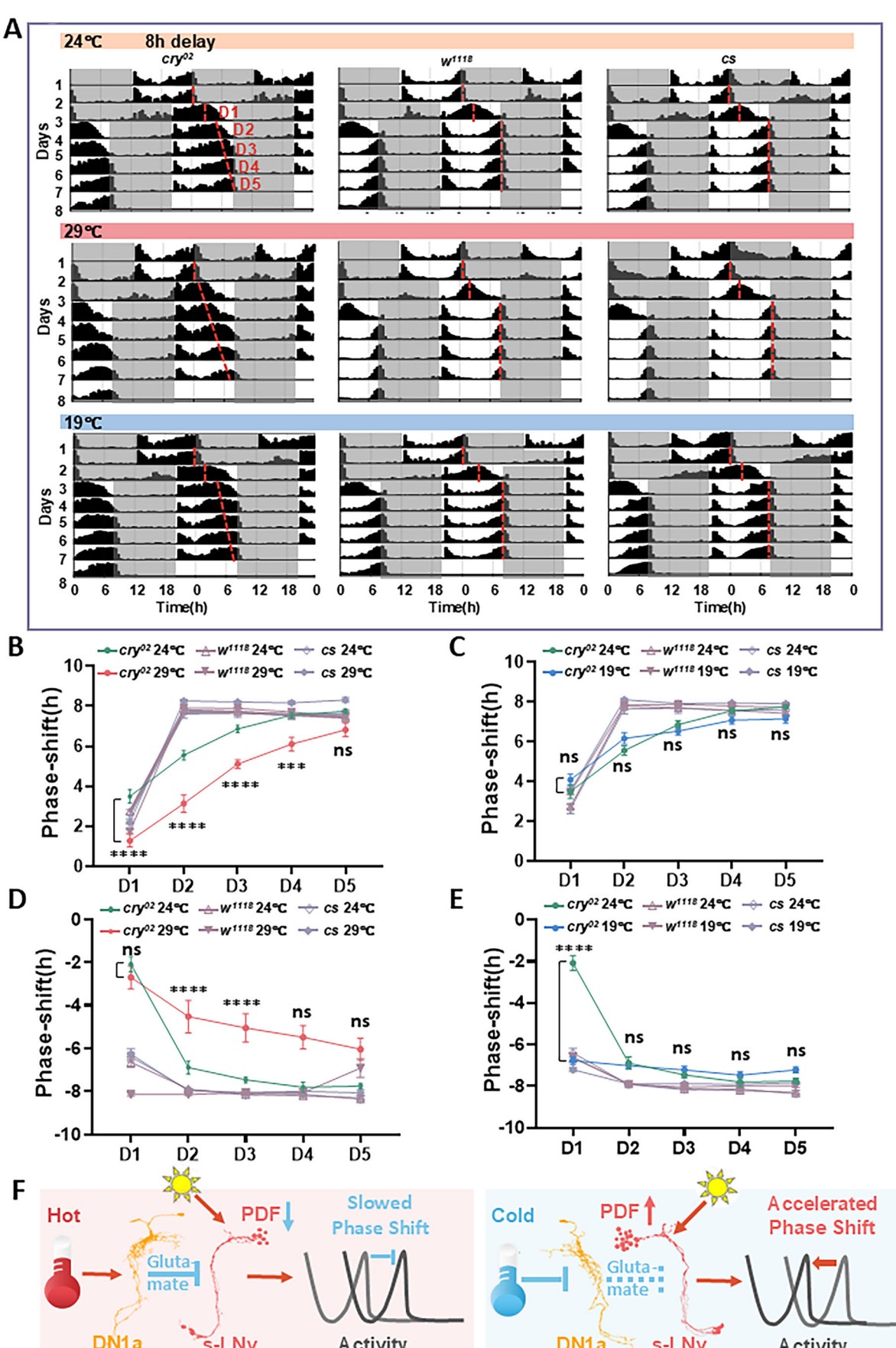

◄

**Figure 7. PDF-mediated light-induced phase shifts of circadian rhythms under different temperatures.**

(A) Activity rhythms of $cry^{02}$ mutant flies and wild-type control flies under 12/12 h LD before and after an 8 h LD phase delay shift at different temperatures. The red dashed line indicates the time of evening activity peak. (B, C) Statistical quantification of the magnitude of the phase shifts on day 1 to day 5 after an 8 h LD phase delay at 24 °C, 29 °C, and 19 °C. D1 indicates the first day of the shift. The number of flies used for the LD shift was $N = 22$ for cs, $N = 24$ for w1118, and $N = 18$ for $cry^{02}$ at 24 °C, $N = 34$ for cs, $N = 40$ for w1118, and $N = 14$ for $cry^{02}$ at 29 °C, $N = 31$ for cs, $N = 32$ for w1118, and $N = 30$ for $cry^{02}$ at 19 °C. $P < 0.0001$ for D1, D2, and D3, $P = 0.0002$ for D4, $P = 0.0082$ for D5 in (B). $P = 0.0355$ for D5 in (C). (D, E) Statistical quantification of the magnitude of the phase shifts on day 1 to day 5 after an 8 h LD phase advance at 24 °C, 29 °C, and 19 °C. The number of flies used for the LD shift was $N = 43$ for cs, $N = 42$ for w1118, and $N = 41$ for $cry^{02}$ at 24 °C, $N = 15$ for cs, $N = 24$ for w1118, and $N = 23$ for $cry^{02}$ at 29 °C, $N = 32$ for cs, $N = 32$ for w1118, and $N = 32$ for $cry^{02}$ at 19 °C. $P = 0.001$ for D2, $P < 0.0001$ for D3, $P = 0.0099$ for D4, $P = 0.0374$ for D5 in (D). $P < 0.0001$ for D1 in (E). (F) A schematic summary of the compensatory mechanisms through which PDF mediates light-induced phase shifts based on temperature. Left: High temperature activates DN1as and then inhibits s-LNvs, which reduces the light-induced firing and PDF release. Consequently, the phase delay shift is slowed down. Right: Under low-temperature conditions, DN1as are inhibited, and the light-induced firing and PDF release of s-LNvs are increased. The phase advance shift is slightly accelerated. Data information: Data are presented as means ± SEM. *$P < 0.05$, **$P < 0.01$, ***$P < 0.001$, ****$P < 0.0001$. Statistical significance was compared using three-way ANOVA with Tukey's post hoc. Source data are available online for this figure.

likely due to either ceiling effects of the GCaMP calcium indicator or the minimal effect of removing already low nocturnal DN1a neuronal activity compared to higher daytime activity. Importantly, we showed that temperature plays a pivotal role in phase adjustment, with lower temperatures (19 °C) accelerating and higher temperatures (29 °C) slowing phase shifts to new light cycles. This temperature-dependent effect on circadian photoentrainment is mediated by s-LNv neurons and operates independently of the CRY pathway. These findings reveal a novel circadian circuit logic through which temperature fine-tunes circadian clock adaptation to light changes, indicating a crucial adaptation for survival in natural environments. The independence of temperature-mediated phase adjustments from CRY not only broadens our understanding of circadian plasticity but also underscores the potential for targeting temperature-sensitive pathways in applications related to circadian phase shift regulation.

The physiological response of central clock neurons to light offers crucial insights into the neural circuits mediating the integration and transmission of light input. Previous studies have reported widespread light responses in circadian pacemaker neurons, including l-LNvs, s-LNvs, the fifth s-LNv, ITP-LNd, DN1a, DN3a, DN3p, and sNPF-LNd neurons, based on patch-clamp recordings (Li et al, 2018). In contrast, our recordings showed that none of the circadian neurons labeled by *CLK856-GAL4* exhibited significant light-induced calcium activity at ZT6. Notably, s-LNvs displayed the most robust light responses at ZT18 (Fig. 2B–D). These discrepancies may stem from the technologies used, as our study employed in vivo two-photon calcium imaging in live flies, providing a more natural physiological context compared to ex vivo electrophysiology. Furthermore, by systematically examining the light responses of s-LNvs at four circadian time points across an entire LD cycle, we revealed that these responses exhibit robust circadian rhythmicity. Thus, our work offers a near-comprehensive characterization of light-responsive pacemaker neurons in the *Drosophila* brain under physiological conditions.

In the *Drosophila* circadian network, several dorsal neuron (DN) subsets, including DN1as, some DN1ps, and DN3s, are glutamatergic and play a role in modulating light response (de Azevedo et al, 2020; Guo et al, 2016; Hamasaka et al, 2007; Hamasaka et al, 2005; Li et al, 2024; Murad et al, 2007; Picot et al, 2007), and their signaling may differentially modulate the circadian photosensitivity of s-LNvs and circadian photoentrainment. For example, disruption of glutamate signaling in DN1ps and DN3s, such as by

downregulating glutamate synthetase, increases rhythmicity under constant light (LL), suggesting reduced photosensitivity (de Azevedo et al, 2020). These findings, together with recent studies showing that a subset of DN1ps and DN3s are inhibited by higher temperatures (Li et al, 2024; Yadlapalli et al, 2018), are consistent with our model in which elevated temperatures reduce s-LNv photosensitivity and slow adaptation to novel light–dark cycles. The precise contribution of different glutamatergic pathways from different circadian neurons to circadian photoentrainment requires further investigation.

In natural environments, animals experience diurnal and seasonal temperature fluctuations. The neural mechanisms by which ambient light and temperature—key environmental zeitgebers—act in synchrony to regulate circadian rhythms remain a critical but underexplored area of research (Green et al, 2015; Menegazzi et al, 2013; Vanin et al, 2012). Our investigation reveals a compensatory mechanism in which key circadian pacemakers calibrate their light-induced firing patterns and circadian peptide release based on temperature variations. High temperatures dampen light-induced pacemaker firing, while low temperatures enhance these responses. This evolutionary strategy likely enables organisms to maintain optimal physiological functions across a range of environmental conditions.

For instance, during hotter periods, reduced light sensitivity in circadian pacemakers may act as an energy conservation strategy. By downregulating light-dependent activities, such as foraging during peak heat, organisms can reduce metabolic demands and lower the risk of overheating (Stevenson et al, 2017). These adaptive mechanisms enable efficient resource allocation, improving survival under fluctuating environmental conditions.

A recent study on *Drosophila sechellia*, a close relative of *Drosophila* melanogaster that inhabits equatorial regions, highlights how species-specific adaptations can arise. *D. sechellia* has lost the ability to delay its evening activity peak under long photoperiods due to minimal seasonal variation in photoperiods near the equator (Shahandeh et al, 2024). This loss is attributed to the species-specific temporal expression of PDF, demonstrating how PDF plasticity contributes to ecological specialization and evolutionary divergence. Such findings underscore the importance of robust compensatory mechanisms in integrating light and temperature cues. Species with greater adaptability to fluctuating environmental conditions may better optimize fitness, influencing interspecies interactions and ecosystem dynamics amid global climate change.

The discovery that temperature modulates light-induced phase adjustments provides valuable insights with practical implications

for human health. Leveraging temperature control in conjunction with light exposure could offer innovative strategies to address circadian misalignment, such as jet lag or shift work. By targeting circadian pacemakers through synergistic interventions involving temperature and light, it may be possible to accelerate phase alignment to new time zones or work schedules. These approaches present a non-invasive, practical solution for enhancing circadian health in modern lifestyles.

# Methods

### Reagents and tools table

| Reagent/resource | Reference or source | Identifier or catalog number |
|---|---|---|
| **Experimental models** | | |
| *D. melanogaster: Clk856-GAL4* | Bloomington Drosophila Stock Center | RRID: BDSC_93198 |
| *D. melanogaster: SS00681* | Bloomington Drosophila Stock Center | RRID: BDSC_87860 |
| *D. melanogaster:UAS-GCaMP6s* | Bloomington Drosophila Stock Center | RRID: BDSC_42746 |
| *D. melanogaster: UAS-GCaMP7s* | Bloomington Drosophila Stock Center | RRID: BDSC_79032 |
| *D. melanogaster: dvpdf-LexA* | Jae H Park Lab | FlyBase ID: FBtp0135022 |
| *D. melanogaster: LexAop-GCaMP6s* | Bloomington Drosophila Stock Center | RRID: BDSC_44589 |
| *D. melanogaster: norpA* | Bloomington Drosophila Stock Center | RRID: BDSC_9048 |
| *D. melanogaster: UAS-hid* | John R. Nambu Lab | N/A |
| *D. melanogaster: Rh6-GAL4* | Bloomington Drosophila Stock Center | RRID: BDSC_7464 |
| *D. melanogaster: cry$^{02}$* | Bloomington Drosophila Stock Center | RRID: BDSC_86267 |
| *D. melanogaster: SS00675* | Gerald M. Rubin Lab | N/A |
| *D. melanogaster: han$^{5304}$* | Bloomington Drosophila Stock Center | RRID: BDSC_33068 |
| *D. melanogaster: per$^{01}$; UAS-per* | Bloomington Drosophila Stock Center | RRID: BDSC_80684 |
| *D. melanogaster: R14F03-GAL4.AD* | Bloomington Drosophila Stock Center | RRID: BDSC_70551 |
| *D. melanogaster: VT002963-GAL4.DBD* | Bloomington Drosophila Stock Center | RRID: BDSC_73764 |

| Reagent/resource | Reference or source | Identifier or catalog number |
|---|---|---|
| *D. melanogaster: UAS-CsChrimson-tdtomato* | Michael A. Crickmore Lab | N/A |
| *D. melanogaster: LexAop-GCaMP7s* | Bloomington Drosophila Stock Center | RRID: BDSC_80913 |
| *D. melanogaster: UAS-VGLUT RNAi* | Vienna Drosophila Resource Center | RRID: VDRC_2574 |
| *D. melanogaster: w$^{1118}$* | Bloomington Drosophila Stock Center | RRID: BDSC_6326 |
| *D. melanogaster: CantonS* | Bloomington Drosophila Stock Center | RRID: BDSC_64349 |
| **Antibodies** | | |
| Rabbit anti-DsRed | Takara Bio USA | Cat# 632496 |
| Chicken anti-GFP | Abcam | RRID: AB_300798 |
| Goat anti-Chicken IgY (H + L), Alexa Fluor 488 | Thermo Fisher Scientific | Cat# A-11039 |
| Goat anti-Rabbit IgG (H + L) Alexa Fluor 635 | Thermo Fisher Scientific | Cat# A-31577 |
| **Chemicals, enzymes, and other reagents** | | |
| Sodium chloride | Sinopharm Chemical Reagent Co., Ltd | Cat#10019318 |
| Potassium chloride | Sinopharm Chemical Reagent Co., Ltd | Cat#10016318 |
| TES | Sigma-Aldrich | Cat#T1375 |
| Trehalose | Sinopharm Chemical Reagent Co., Ltd | Cat#63012666 |
| Glucose | Sinopharm Chemical Reagent Co., Ltd | Cat#G8270 |
| Sodium hydrogen carbonate | Sinopharm Chemical Reagent Co., Ltd | Cat#10018960 |
| Sodium dihydrogen phosphate | Sinopharm Chemical Reagent Co., Ltd | Cat#20040818 |
| Calcium chloride anhydrous | Sinopharm Chemical Reagent Co., Ltd | Cat#10005861 |
| Magnesium chloride anhydrous | Sinopharm Chemical Reagent Co., Ltd | Cat#7786-30-3 |
| Sodium hydroxide | Sinopharm Chemical Reagent Co., Ltd | Cat#10019719 |
| Hydrogen chloride | Sinopharm Chemical Reagent Co., Ltd | Cat#10011018 |
| All-trans-retinal (ATR) | Sigma-Aldrich | RRID:R2500-1G |

| Reagent/resource | Reference or source | Identifier or catalog number |
|---|---|---|
| **Software** | | |
| Fiji/ImageJ | https://imagej.net/ | RRID:SCR_003070 |
| MATLAB | MathWorks | RRID: SCR_001622 |
| Prism GraphPad | http://graphpad.com/ | RRID: SCR_000306 |
| pySolo | https://www.pysolo.net/ | N/A |
| The MATLAB scripts for SCAMP | Leslie C Griffith Lab | https://academics.skidmore.edu/blogs/cvecsey/?page_id=57 |
| The MATLAB scripts for the phase shift | Jeffrey C Hall Lab | N/A |

## Fly strains

Flies were reared on standard medium under a 12-h light to 12-h dark cycle at 25 °C with a humidity level of 60% unless otherwise noted. The following lines were used to generate the transgenic flies used in this study: *clk856-GAL4* (Yao et al, 2012), *UAS-GCaMP6s* (Bloomington No. 42746), *s-LNv* (GMR-SS00681), *UAS-GCaMP7s* (Bloomington No. 79032), *dvpdf-LexA* (Bahn et al, 2009), *LexAop-GCaMP6s* (Bloomington No. 44589), *norpA* (Bloomington No. 9048), *UAS-hid* (Wing et al, 2001), *Rh6-GAL4* (Bloomington No. 7464), *cry[02]* (Bloomington No. 86267), *l-LNv* (GMR-SS00675), *han[5304]* (Bloomington No. 33068), *per[01]; UAS-per* (Bloomington No. 80684), *DN1a (R14F03-GAL4.AD; VT002963-GAL4.DBD)*, *UAS-CsChrimson-tdtomato* (Zhang et al, 2019), *LexAop-GCaMP7s* (Bloomington No. 80913), *UAS-VGLUT RNAi* (VDRC No. 2574), *w[1118]*, *CantonS*.

## Peltier setup

The temperature of the Peltier element was measured using a thermistor, and a closed-loop feedback system employing proportional-integral-derivative control was utilized to regulate it. The temperature control device (SLD70) was purchased from Sunny Precise Instruments (Shanghai) Co., Ltd. For the two-photon calcium imaging of live flies that experienced cooling and heating, the fly brain was submerged in saline. The temperature between the saline solution and the Peltier element was closely alignment due to the small thermal resistance. Moreover, the time and amplitude of the cooling or heating during all experiments are uniform.

## ATR feeding

The 100 mM stock solution of all-trans-retinal (ATR) (Sigma) was prepared by dissolving it in 100% alcohol. The final concentration of ATR food utilized in optogenetic coupled two-photon calcium imaging was 800 μ, with 240 μl of the stock solution mixed with 30 ml of 5% sucrose and 1% agar medium. 3 ~ 5 days flies were transferred to ATR food for 3 days before conducting surgery.

## Two-photon calcium imaging in vivo

Calcium imaging in vivo was performed following a previously published procedure (2013-Li). Flies aged 2 ~ 5 days were maintained for three days under a 12:12 LD cycle at 24 °C. On the fourth day, the flies at a specific time window were glued onto a custom-made recording chamber, with the antenna pointing below the chamber exposed to air. In detail, for experiments during the ZT0-12 time points, flies were removed from the LD entrainment incubator 30 min prior to imaging, and surgery to create a cuticle window was performed under normal lighting conditions. Flies were then placed under the two-photon microscope and allowed to acclimate for at least 5 min before the start of the experiment. It's important to note that two-photon microscopy imaging requires protection from light, so the environment during the actual imaging period was effectively dark at all time points except for the light pulse period.

Specifically, for the ZT12 time point, flies were removed at ZT11.5, and surgery was performed under normal light. At the ZT12 light-off time point, flies were transferred to the two-photon microscope for imaging. For experiments during ZT12-24, flies were removed from the incubator under dark conditions and all surgical procedures were performed under dim red light to avoid circadian disruption. After surgery, the flies were imaged under the two-photon microscope.

In particular, due to the different projection patterns of clock neurons and the limitations of the cuticle window during two-photon imaging, we had to monitor different neuronal compartments in different cell types. Some neurons can be identified and imaged at the cell body, while others, such as LNds, were only accessible at the processes. This compartmental difference is a limitation when comparing response patterns between cell bodies and neuronal projections, as the baseline calcium dynamics may differ between these compartments.

For the experiments imaging l-LNvs, we positioned the fly head at a steeper angle and created an imaging window between the ocelli and the antennae to better visualize the l-LNv cell bodies. The imaging was performed at the anterior region of the fruit fly brain, specifically recording the cell bodies of l-LNvs rather than their projections.

The brain was exposed and immersed in recording solution (103 mM 345 NaCl, 3 mM KCl, 5 mM TES, 8 mM trehalose, 10 mM glucose, 26 mM NaHCO$_3$, 1 mM 346 NaH$_2$PO$_4$, 1.5 mM CaCl$_2$, and 4 mM MgCl$_2$, osmolarity adjusted to 280–300 mOsm, pH adjusted to 7.3).

Olympus FV1200MPE microscope with a 25×, N.A. = 1.1 Water-immersion objective was used for calcium imaging. The excitation laser for GCaMP was 920 nm of the Spectra-Physics Mai Tai® ultrafast laser. After the dorsal neurons were located, a brain volume of about 200 μm × 200 μm × 30 μm through time-lapsed Z-series with 6 optical layers was captured. Notably, the brain volume captured in cooling and heating experiments was about 200 μm × 200 μm × 90 μm with 15 optical layers. The captured image of a single layer has a pixel size of 0.414 μm, spatial resolution of 512 × 512, and temporal resolution of 0.9 Hz. The wavelengths of the light stimuli during two-photon calcium imaging are precisely controlled: blue (460–462 nm), green (522–525 nm), and yellow (570–572 nm).

## Calcium imaging data analysis

The raw image stacks were first registered in ImageJ with the TurboReg plugin (Thevenaz et al, 1998) to correct for brain motion. If there was too much movement to obtain a stable image stack after registration, the data was abandoned. The image stacks were then analyzed using custom programs written with MATLAB. Regions of interest (ROIs) were manually drawn around either somata or fibers of target neurons. In detail, for flies expressing GCaMP driven by *CLK856-GAL4*, we performed volumetric two-photon microscopy imaging focused on the dorsal brain region. Importantly, our analysis protocol allowed us to reliably distinguish between different clock neuron populations through careful 3D reconstruction of the brain using FIJI.

The cell bodies of DN1a, DN1p, and DN2 neurons were identified based on their distinct locations within the Z-stack, while s-LNv and LNd neurons were identified based on their characteristic fiber patterns. This approach takes advantage of several anatomical features that allow reliable differentiation: DN1a neurons (two per hemisphere) have cell bodies positioned well above the dorsal fibers and separated from DN1ps. DN1p neurons (several per hemisphere) are clustered near the dorsal fibers, either overlapping or slightly above them. DN2 neurons (two per hemisphere) have cell bodies close to, but below, the dorsal fibers.

Notably, for heat maps in Figures, each row represents a ROI. For each brain sample analyzed, we typically defined two ROIs—one from the left hemisphere and one from the right hemisphere. Each brain preparation corresponds to one independent experiment. This experimental design allows us to capture bilateral responses while maintaining the identity of individual experimental replicates in our heatmap visualizations.

To determine $\Delta F/F$ for each ROI, the baseline fluorescence level before light pulse onset was averaged and then subtracted and divided from the trace. Peak $\Delta F/F$ ($\Delta F/F_{max}$) during the light pulse was measured as the amplitude of the light-induced response.

## Immunostaining

Adult flies were dissected in the PBS, then fixed for 20 min at room temperature in PBST (PBS + 0.5% Triton-X) with 4% paraformaldehyde. After washing six times using PBST, samples were incubated with primary antibodies at 4 °C overnight. Brains were washed six times in PBST and incubated in secondary antibodies for two hours at room temperature, then washed six times in PBST. After that, samples were mounted in Vectashield Mounting Medium (Vector Laboratories) and viewed in 2-μm sections on a laser scanning confocal microscope (Olympus) under ×20 magnification. Primary antibodies used in this paper were: rabbit anti-DsRed (Takara, Cat# 632496, 1:200), chicken anti-GFP (Abcam, RRID AB_300798, 1:1000).

## Sleep recording and data analysis

In all, 3–6-day-old male flies were used to record locomotor activity rhythms. Flies were confined in white 96-well Microfluor 2 plates (Fishier) containing 300ul food (5% sucrose and 1% agar) in a previously reported Flybox system (Guo et al, 2017). The fly movement was monitored using a webcam with an interval of 10 s. The Flybox was placed in an incubator with a constant humidity of 65%. Meanwhile, the temperature in the incubator can be controlled precisely. White LEDs were set in Flybox, and lights on and off were controlled by a timing socket.

To ensure that all fly strains were entrained completely under 12/12 h LD, the locomotor activity of all strains was first recorded for 3 d under 12/12 h LD for phase shifts of 24 °C and 19 °C, 2 d for 29 °C. Notably, for the phase shift 29 °C, all fly strains were entrained to the 12/12 h LD cycles for 3 days at 29 °C before being loaded in Flybox. Subsequently, the flies were subjected to either an 8 h phase advance by a cut down of the light phase or an 8 h phase delay of LD by prolonging the light phase. At least twice, each of the tests was repeated. The raw data of locomotor activity were analyzed using MATLAB program (SCAMP) (Donelson et al, 2012; Gilestro and Cirelli, 2009). To calculate the magnitude of the phase shift, daily activity profiles of individual flies were plotted, and the phase of the evening peak of each fly was determined manually, as described previously (Levine et al, 2002). Specifically, the raw data were smoothed with a digital low-pass filter, and the cutoff frequency was set to 6 h. This level of smoothing enhanced the clarity of the activity profile while preserving the distinction between the lights-off and evening peaks. The time of the evening peak was then determined by manually removing subpeaks using the mouse pointer in MATLAB. The phase angles between the evening peak before the day of the LD phase shift and the evening peak on day 1 to day 5 after the LD phase shift were calculated for individual flies. The mean phase angles were calculated for each strain. For statistical analysis, Tukey's multiple comparison test was used. Statistics were calculated using GraphPad Prism software.

## Statistical analysis

Statistical analysis was performed using GraphPad Prism software. No statistical tests were used to predetermine sample sizes. All measurements were taken from distinct samples unless otherwise stated. Pairwise comparison between all groups was performed by one-way ANOVA, followed by Tukey's post hoc in Figs. 2–6. Comparison between selected groups was performed by two-tailed unpaired Student's t test with post hoc Benjamini–Hochberg FDR correction in Figs. 3, 4, and 6. Pairwise comparison between all groups was performed by three-way ANOVA, followed by Tukey's post hoc in Fig. 7. The number of flies (*n* value) was indicated in the figure legends. The responses were presented as mean values, and error bars represented the standard error of the mean (SEM). Statistical significance was determined with an asterisk if the *P* value was less than 0.05 ($P < 0.05$), two asterisks for $P < 0.01$, three asterisks for $P < 0.001$, and four asterisks for $P < 0.0001$. No data were excluded from the analysis.

Complete blinding was not feasible in this study as the experimental design required specific genotypes for the experimental and control groups. However, to minimize potential bias, the data analysis was blinded to group allocation when assessing outcomes.

# Data availability

This study includes no data deposited in external repositories. The source data of this paper are collected in the following database record: https://www.ebi.ac.uk/biostudies/studies/S-BSST2021.

The source data of this paper are collected in the following database record: biostudies:S-SCDT-10_1038-S44318-025-00499-w.

## Peer review information

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

## Acknowledgements

The authors thank Dr. Zhefeng Gong, Dr. Chang Liu, Dr. Yi Rao, Dr. Yufeng Pan, Dr. Junhai Han, Dr. Wei Zhang, the Vienna Drosophila Resource Center and the Bloomington Drosophila Stock Center for stocks and reagents. The authors are grateful to Shuangshuang Liu for the technical support in the Core Facilities, Zhejiang University School of Medicine. The authors are grateful to Sanhua Fang and Dan Yang for the technical support in the Core Facilities, Zhejiang University School of Brain Science and Brain Medicine. This work is supported by funding from the National Natural Science Foundation of China (32171008 and 32471210), the Non-profit Central Research Institute Fund of Chinese Academy of Medical Sciences (2023-PT310-01), and the Fundamental Research Funds for the Central Universities (2025ZFJH01-01 and 226-2024-00133) to FG.

## Author contributions

**Yue Tian**: Data curation; Formal analysis; Investigation; Methodology; Writing—original draft; Writing—review and editing. **Hailiang Li**: Resources; Software; Formal analysis; Methodology. **Wenjing Ye**: Resources; Software; Methodology. **Xin Yuan**: Resources; Software; Methodology. **Xuan Guo**: Resources. **Fang Guo**: Conceptualization; Formal analysis; Supervision; Funding acquisition; Investigation; Writing—original draft; Project administration; Writing—review and editing.

Source data underlying figure panels in this paper may have individual authorship assigned. Where available, figure panel/source data authorship is listed in the following database record: biostudies:S-SCDT-10_1038-S44318-025-00499-w.

## Disclosure and competing interests statement

The authors declare no competing interests.

# Expanded View Figures

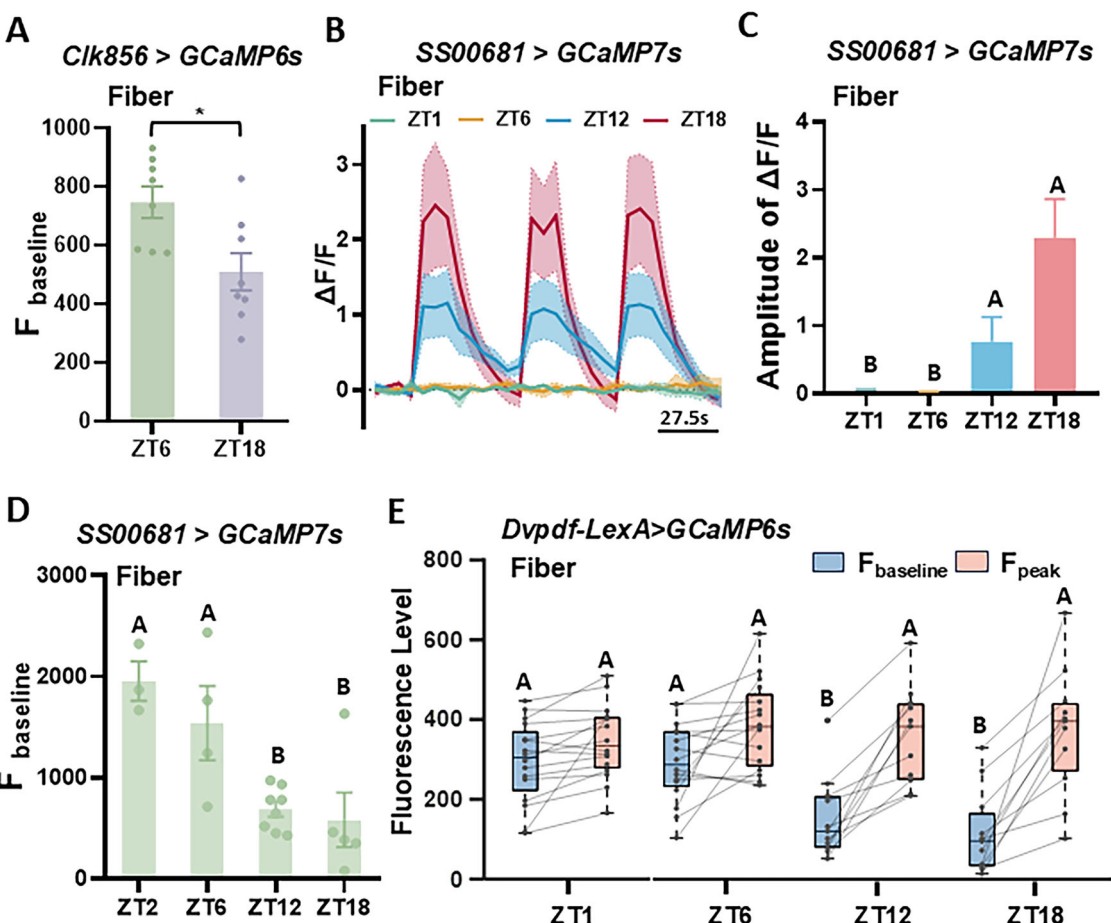

**Figure EV1. Basal calcium activity level.**

(A) Basal calcium activity level of circadian neuron dorsal fibers expressing GCaMP labeled by *CLK856-GAL4*. $N = 4$ for both ZT6 and ZT18. $P = 0.0127$. (B) Traces of $\Delta F/F_0$ of s-LNvs labeled by a split-GAL4 driver at different time points in response to light pulse. (C) Statistical quantification of the response amplitudes of s-LNvs at different time points. $N = 3$ for ZT1, ZT6, and ZT18, and $N = 4$ for ZT12. (D) Basal calcium activity level of the s-LNvs dorsal fibers labeled by specific split-gal4. $N = 3$ for ZT1, ZT6, and ZT18, and $N = 4$ for ZT12. (E) The basal calcium fluorescence $F_{baseline}$ (blue) and maximum calcium fluorescence $F_{peak}$ (pink) of s-LNv dorsal fibers at different time points in wild-type control flies. Data information: Data are presented as means ± SEM. $*P < 0.05$. Analyses of two samples employed a two-tailed unpaired Student's *t* test for A. One-way ANOVA with the Tukey correction to the $P$ value was used to assess the significance for multiple samples in (C, D, E). The letters A and B above or below the curves denote significantly different means within each of the two groups, $P < 0.05$.

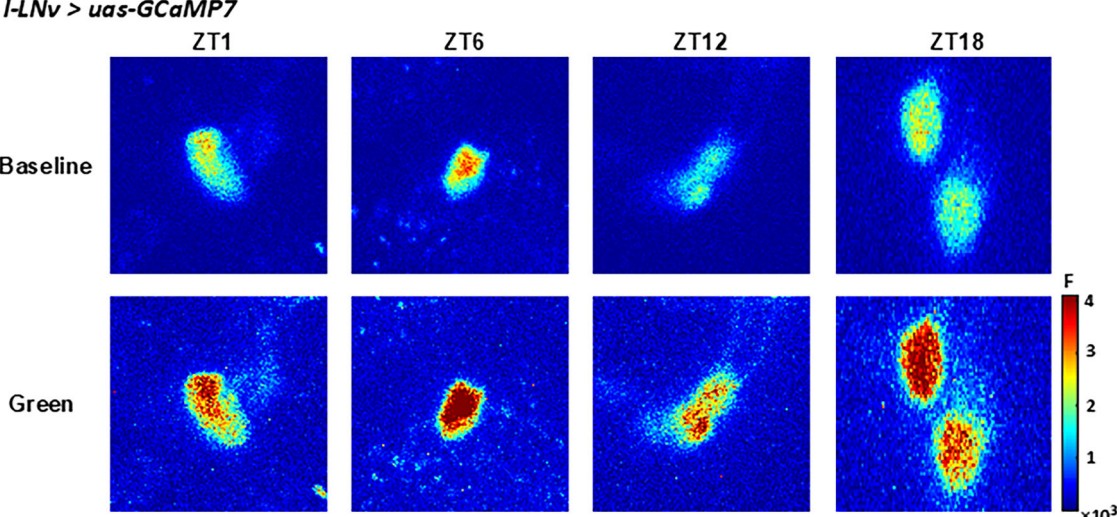

**Figure EV2. Light response of l-LNv soma.**

From left to right panels: Pseudocolor maps of the basal calcium activity (top row) and calcium activity changes in response to green LED stimulation (bottom row) at ZT1, ZT6, ZT12, and ZT18. Warmer colors indicate higher calcium levels. Color scales are shared across panels.

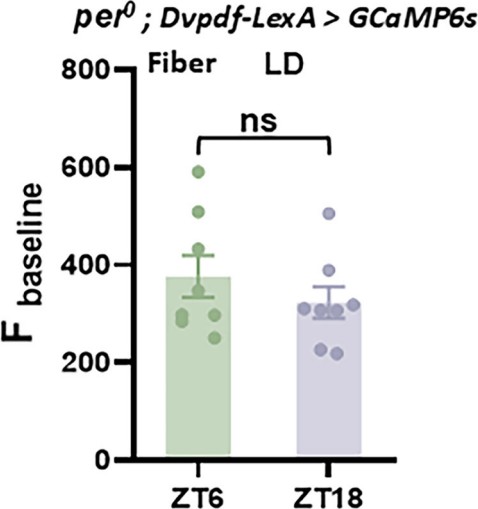

**Figure EV3.  Basal calcium activity level of the s-LNvs dorsal fibers in *per⁰* mutant flies under LD.**

$N = 4$ for both ZT6 and ZT18. Data information: Data are presented as means ± SEM. Analysis of two samples employed a two-tailed unpaired Student's *t* test.

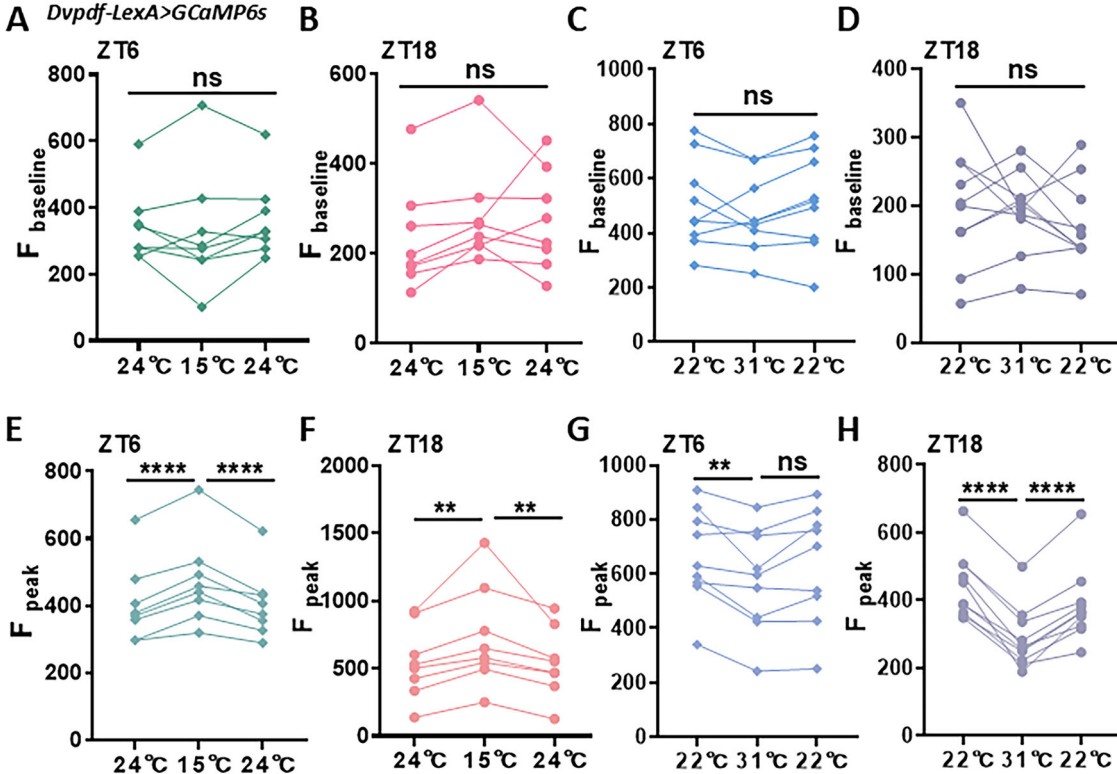

**Figure EV4. Temperature-induced changes of light responses in s-LNvs.**

(A, B) The basal calcium fluorescence of s-LNv dorsal fibers before, during, and after cooling at ZT6 (A) and ZT18 (B), respectively. $N = 6$ in (A), $N = 4$ for in (B). (C, D) The basal calcium fluorescence of s-LNv dorsal fibers before, during, and after heating at ZT6 (C) and ZT18 (D), respectively. $N = 4$ in (C), $N = 7$ in (D). (E, F) The maximum calcium fluorescence of s-LNv dorsal fibers to light pulse before, during, and after cooling at ZT6 (E) and ZT18 (F). $N = 6$ for ZT6, $N = 4$ for ZT18. $P < 0.0001$ for both 24 °C vs 15 °C and 15 °C vs 24 °C in (E). $P = 0.0036$ for 24 °C vs 15 °C, $P = 0.003$ for 15 °C vs 24 °C in (F). (G, H) The maximum calcium fluorescence of s-LNv dorsal fibers to light pulse before, during, and after heating at ZT6 (G) and ZT18 (H). $N = 4$ for ZT6, $N = 7$ for ZT18. $P = 0.0047$ for 22 °C vs 31 °C in (G). $P < 0.0001$ for both 22 °C vs 31 °C and 31 °C vs 22 °C in (H). Data information: Data are presented as means ± SEM. **$P < 0.01$, ****$P < 0.0001$. One-way ANOVA with the Tukey correction to the $P$ value was used.

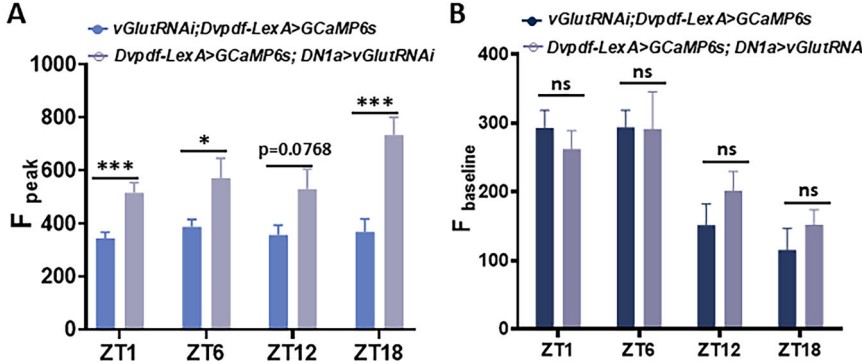

**Figure EV5. DN1as exert glutamatergic inhibition to s-LNvs.**

(A) The maximum calcium fluorescence (right) of s-LNv dorsal fibers to light pulse in wild-type control flies (blue) and DN1a-inhibited flies (purple). The number of wild-type control flies was $N = 8$ for ZT1 and ZT6, $N = 6$ for ZT12 and ZT18. The number of DN1a-inhibited flies was $N = 8$ for ZT1, ZT6, and ZT12, and $N = 6$ for ZT18. $P = 0.0005$ for ZT1, $P = 0.03$ for ZT6, $P = 0.0003$ for ZT18. (B) The basal calcium fluorescence of s-LNv dorsal fibers in wild-type control flies (blue) and DN1a-inhibited flies (purple). The number of wild-type control flies was $N = 8$ for ZT1 and ZT6, $N = 6$ for ZT12 and ZT18. The number of DN1a-inhibited flies was $N = 8$ for ZT1, ZT6, and ZT12, and $N = 6$ for ZT18. Data information: Data are presented as means ± SEM. *$P < 0.05$, **$P < 0.01$, ***$P < 0.001$. Two-tailed unpaired Student's $t$ test was used.

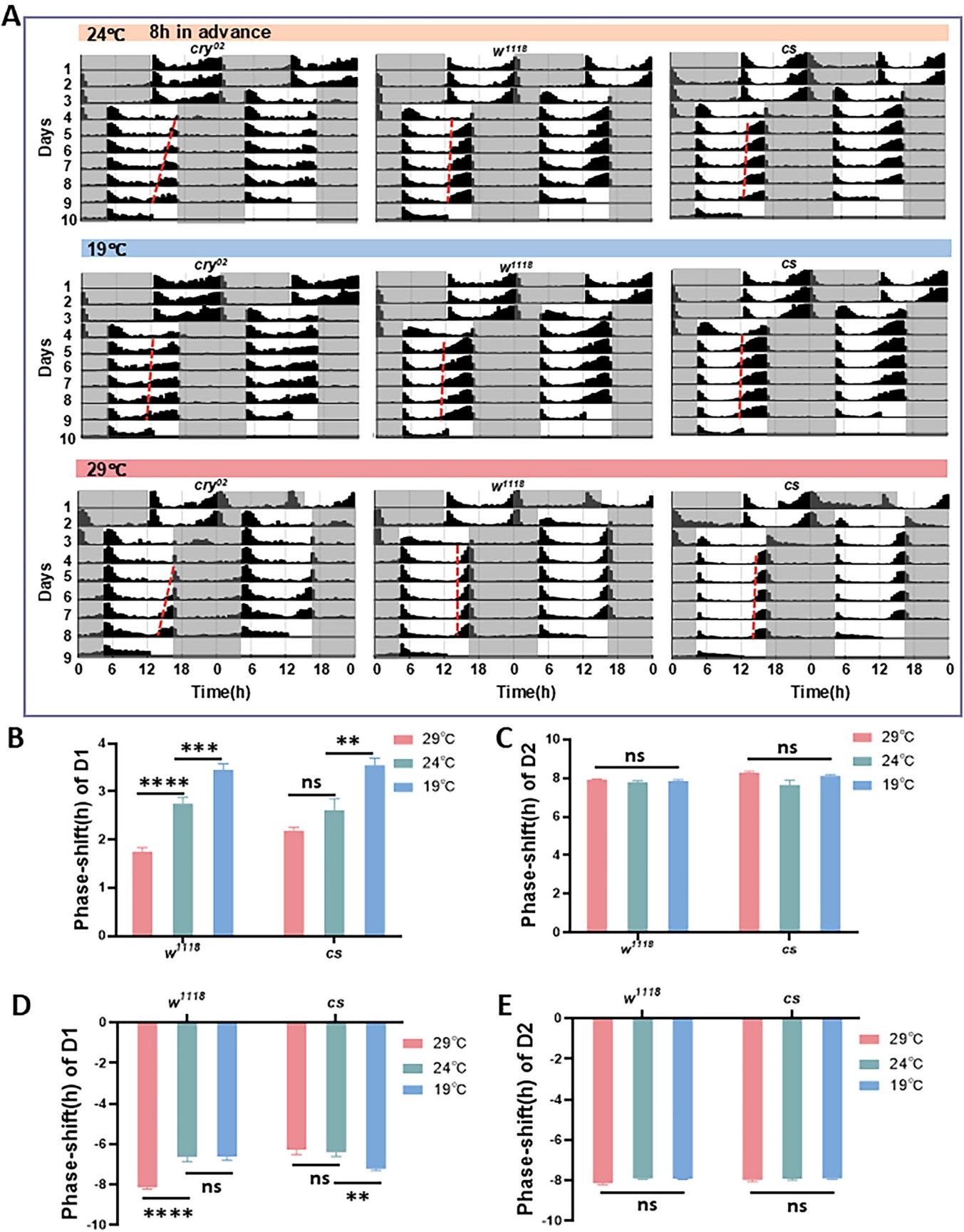

◄ **Figure EV6.  Activity rhythms of other data were not included in Fig. 7.**

(A) Under 12/2 h LD with an 8 h LD phase advance at 24 °C, 19 °C, and 29 °C. The gray area in the actograms indicates the dark phase. The red dashed line indicates the onset time of the evening activity peak. (B, C) Statistical quantification of the phase shift magnitude of wild-type flies on day 1 (B) and day 2 (C) after an 8 h LD phase delay at 24 °C, 29 °C, and 19 °C. The number of $w^{1118}$ flies was $N = 24$ at 24 °C, $N = 40$ at 29 °C, N = 32 at 19 °C. The number of *cs* flies was $N = 22$ at 24 °C, $N = 34$ at 29 °C, $N = 31$ at 19 °C. For $w^{1118}$ flies in (B), $P < 0.0001$ for 29 °C vs 24 °C, $P = 0.0002$ for 19 °C vs 24 °C. For *cs* flies in (B), $P = 0.0011$ for 19 °C vs 24 °C. (D, E) Statistical quantification of the phase shift magnitude of wild-type flies on day 2 (D) and day 2 (E) after an 8 h LD phase delay at 24 °C, 29 °C, and 19 °C. The number of $w^{1118}$ flies was $N = 42$ at 24 °C, $N = 24$ at 29 °C, $N = 32$ at 19 °C. The number of *cs* flies was $N = 43$ at 24 °C, $N = 15$ at 29 °C, $N = 32$ at 19 °C. For $w^{1118}$ flies in (D), $P < 0.0001$ for 29 °C vs 24 °C. For *cs* flies in (D), $P = 0.0085$ for 19 °C vs 24 °C. Data information: Data are presented as means ± SEM. $**P < 0.01$, $***P < 0.001$, $****P < 0.0001$. One-way ANOVA with the Tukey correction to the P value was used to assess the significance.

