## [Peer Review File · The EMBO Journal]

Temperature-dependent modulation of light-induced circadian responses in *Drosophila melanogaster*

Yue Tian, Hailiang Li, Wenjing Ye, Xin Yuan, Xuan Guo, and Fang Guo

Corresponding author(s): Fang Guo (gfang@zju.edu.cn)

Review Timeline:

Submission Date:	23rd Dec 24
Editorial Decision:	31st Jan 25
Revision Received:	24th Mar 25
Editorial Decision:	27th Apr 25
Revision Received:	14th May 25
Editorial Decision:	2nd Jun 25
Revision Received:	4th Jun 25
Accepted:	5th Jun 25

Editor: Ioannis Papaioannou

Transaction Report:

Dear Prof. Guo,

Thank you for submitting your manuscript EMBOJ-2024-120009 for consideration by The EMBO Journal, and for your patience during peer review. Your manuscript has now been seen by three experts in the field, and we have received the full set of their well-informed and detailed reports, which are included below.

As you will see, the feedback of all three referees on your study is very supportive. They all find the manuscript interesting and addressing an important question in the field. They also point out that the experimental work is of high quality and the manuscript well-prepared. The referees also identify some limitations and provide helpful and reasonable suggestions for further improvement of the study and the manuscript, which would increase the impact of the work on the field.

Given the referees' positive comments and recommendations, I would like to invite you to submit a revised version of the manuscript along with a detailed point-by-point response addressing all referees' comments. I should add that it is The EMBO Journal policy to allow only a single round of major revision, and acceptance of your manuscript will therefore depend on the completeness of your responses in this revised version. Please let me know if you have any questions or comments that you would like to discuss with me.

We generally allow three months as standard revision time (April 30, 2025). As a matter of policy, competing manuscripts published during this period will not negatively impact our assessment of the conceptual advance presented by your study. However, we request that you contact us as soon as possible upon publication of any related work, to discuss how to proceed. Should you foresee a problem in meeting this three-month deadline, please let us know in advance and we may be able to grant an extension.

Thank you for the opportunity to consider your work for publication in The EMBO Journal. I look forward to your revision.

Best regards,

Ioannis

Instructions for preparing your revised manuscript

1. When you are ready to submit the revision, please upload:

- A Word file of the manuscript text (including legends of main Figures, EV Figures and Tables). Please make sure that changes are highlighted (or "tracked") to be clearly visible.

- Individual production-quality figure files (one file per figure). When assembling your figures, please refer to our figure preparation guidelines in order to ensure proper formatting and readability in print as well as on screen:

If the data shown in a figure are obtained from n {less than or equal to} 2, please use scatter plots showing the individual data points.

- i. the name of the statistical test used to generate error bars and P values
- ii. the number (n) of independent experiments (please specify technical or biological replicates) underlying each data point (discussion of statistical methodology can be reported in the Materials and Methods section, but figure legends should contain a basic description of n , P , and the test applied)
- iii. the nature of the bars and error bars (s.d., s.e.m.).

- A point-by-point response to the referees' comments, with a detailed description of the changes made (as a word file). All referees' concerns must be fully addressed and their suggestions taken on board. When preparing your letter of response to the referees' comments, please bear in mind that this will form part of the Review Process File and will therefore be available online

to the community. Please note that you have the possibility to opt out of the transparent process at any stage prior to publication by letting the editorial office know (contact@embojournal.org); if you do opt out, the Review Process File link will point to the following statement: "No Review Process File is available with this article, as the authors have chosen not to make the review process public in this case.". For more details on our Transparent Editorial Process, please visit our website: <https://www.embopress.org/page/journal/14602075/authorguide#transparentprocess>

- Expanded View (EV) files (replacing Supplementary Information) that are collapsible/expandable online. A maximum of 5 EV Figures can be typeset. EV Figures should be cited as "Figure EV1, Figure EV2" etc. in the text, and their respective legends should be included in the manuscript file after the legends of regular figures. See detailed instructions regarding Expanded View files here:

- For the figures that you do NOT wish to display as Expanded View figures, they should be bundled together with their legends in a single PDF file called "Appendix", which should start with a short Table of Contents (including page numbers). Appendix figures should be referred to in the main text as: "Appendix Figure S1, Appendix Figure S2" etc. Please see detailed instructions here: <https://www.embopress.org/page/journal/14602075/authorguide#expandedview>

- A complete author checklist, which you can download from our author guidelines (<https://www.embopress.org/page/journal/14602075/authorguide>). Please note that the checklist will also be part of the Review Process File.

2. Please note that no statistics should be calculated and shown in Figures if $n=2$. Please also note that each p value should be reported as an exact value.

3. Before submitting your revision, primary datasets (and computer code, where appropriate) produced in this study need to be deposited in appropriate public databases (see <https://www.embopress.org/page/journal/14602075/authorguide#dataavailability>). The accession numbers, database, and the specific URLs (links) should be listed in a formal "Data availability" section (placed after Methods), following the example below:

"The RNA-seq datasets produced in this study are available in the following database:
Gene Expression Omnibus GSE46843 (<https://www.ncbi.nlm.nih.gov/geo/query/acc.cgi?acc=GSE46843>)"

*** All links should resolve to a page where the data can be accessed. ***

*** Please remember to provide in the Data availability section of your revised manuscript reviewer passwords if the datasets are not yet public. ***

*** The Data Availability Section is restricted to new primary data that are part of this study. In case you have no data that require deposition in a public database, please state so instead of referring to the database: "Our study includes no data deposited in public repositories." under the heading "Data availability". ***

4. Please check that the title and the abstract of the manuscript are brief, yet explicit, even to non-specialists. The length of the title should not exceed 100 characters, and the abstract should be a single paragraph not exceeding 175 words.

5. Please also note our reference format: <https://www.embopress.org/page/journal/14602075/authorguide#referencesformat>.

7. Please remember: digital image enhancement is acceptable practice, as long as it accurately represents the original data and conforms to community standards. If a figure has been subjected to significant electronic manipulation, this must be noted in the figure legend or in the "Materials and Methods" section. The editors reserve the right to request original versions of figures and the original images that were used to assemble the figure.

8. Our journal encourages inclusion of data citations in the reference list to directly cite datasets that were obtained from public databases. Data citations in the article text are distinct from normal bibliographical citations and should directly link to the database records from which the data can be accessed. In the main text, data citations are formatted as follows: "Data ref: Smith et al, 2001" or "Data ref: NCBI Sequence Read Archive PRJNA342805, 2017". In the Reference list, data citations must be labeled with "[DATASET]". A data reference must provide the database name, accession number/identifiers, and a resolvable link to the landing page from which the data can be accessed at the end of the reference. Further instructions are available at: <https://www.embopress.org/page/journal/14602075/authorguide#referencesformat>.

9. We request authors to consider both actual and perceived competing interests. Please review our policy

(<https://www.embopress.org/page/journal/14602075/authorguide#conflictsofinterest>) and update your competing interests statement if necessary. Please name this section 'Disclosure and competing interests statement' and place it after the Acknowledgements section.

10. Please note that all corresponding authors are required to provide an ORCID ID upon submission of a revised manuscript (<https://orcid.org/>). Please find instructions on how to link your ORCID ID to your account in our manuscript tracking system in our Author guidelines (<https://www.embopress.org/page/journal/14602075/authorguide#authorshipguidelines>).

11. We use CRediT to specify the contributions of each author in the journal submission system. CRediT replaces the author contribution section, which should be removed from the manuscript. Please use the free text box to provide more detailed descriptions. See also guide to authors: <https://www.embopress.org/page/journal/14602075/authorguide#authorshipguidelines>.

13. We would also welcome the submission of cover suggestions or motifs to be used by our Graphics Illustrator in designing a cover.

14. Please use the link below to submit your revision:
<https://emboj.msubmit.net/cgi-bin/main.plex>

Referee #1:

This manuscript from Tian et al. assesses the interaction between light and temperature in circadian clock entrainment. Much of the data are derived from 2-photon imaging of GCaMP signals in circadian clock neurons. Although previous studies have independently assessed light- or temperature-evoked responses in clock neurons, this manuscript is unique in that investigates the interaction between these two parameters; namely, how temperature affects clock neuron responses to acute light pulses. The authors first suggest that light responsiveness of the sLNv clock neurons varies across the day, with larger responses during the dark period in LD conditions (or subjective night in DD conditions). As the sLNVs have previously been implicated in generating behavioral rhythms and in mediating light entrainment, these data offer a physiological correlate of these functional attributes. They then assess clock cell responsiveness to light pulses at different ambient temperature and argue that light responses are enhanced at lower temperatures. They attribute the temperature modulation of sLNv light responses to inputs from temperature-sensitive DN1a clock cells, as DN1a ablation potentiates sLNv responses in a manner that mimics the effect of low temperature. Finally, they perform behavioral studies to show that re-entrainment to phase shifts in light cycles is modified by temperature. At lower temperatures, there is a slight acceleration of behavioral re-entrainment. Overall this is a very interesting paper that applies novel methods to address an important question in the field. However, there are a few major issues that need to be addressed, as detailed below.

Major Concerns

1. There are necessary details lacking from the experimental methods section that makes it difficult to adequately assess the merits of the manuscript.
-for the calcium imaging, what were the light conditions during the imaging? Were lights on for ZT0-12 time points? This is important to clarify because light conditions during imaging could alter baseline neuron firing and because ongoing ambient light could lead to activation light-response mechanisms, thereby affecting the acute response to the light pulse.
-in several experiments, the authors have used the Clk856-GAL4 line to drive GCaMP expression in most clock neurons. They have also conducted the majority of their calcium imaging analysis on the dorsal projections of the clock neurons rather than the cell bodies. It is therefore unclear how they are able to differentiate between the different clock cell populations for their quantification (eg in Figure 1C-E). This needs to be thoroughly explained and validated, because the authors claim that light responses are predominantly occurring in the sLNvs compared to other clock neurons. It would seem nearly impossible to differentiate the different individual clock cell populations when imaging from the densely intermingled dorsal projections.
-for experiments in ILNvs (Figure 4A-B), what part of the ILNvs is being imaged? These cells do not project to the dorsal brain so it can't be the same method as used for the other cell types.

2. In the text, the authors say that they used Dvpdf-GAL4, "a driver that can specifically label the dorsal projection fiber of sLNvs". What is the evidence that Dvpdf-GAL4 specifically labels the sLNv projections? The line has previously been described by the same authors are being expressed in both sLNv and LNd cells. The description in the text here is misleading, as they cite Figure 2A in support of the specificity of the driver. Figure 2A is a reconstruction of the sLNv cells from the FlyWire dataset. It would be better to show imaging of the actual Dvpdf-GAL4 expression pattern. Why not use the sLNv-specific split-GAL4 line

(SS00681) that they use in Figure EV1, as this is a cleaner driver that is specific for sLNvs? Furthermore, in Figure 2, it is labeled as Dvpdf-LexA. Could the authors clarify if they have used the GAL4 or LexA line?

3. I am confused about the authors' interpretation of the major findings of the paper. In the abstract and throughout the text, they say that higher temperatures suppress and lower temperatures enhance the responsiveness of sLNv neurons to light. However, later in the results, they say "We hypothesized that the circadian rhythmicity in s-LNvs light responses might arise from the endogenous circadian clock modulating basal calcium levels across day and night. This predicts that higher basal calcium levels during daytime could limit light-induced activation through a ceiling effect, while lower nighttime levels could permit larger light-induced calcium increases." They then go on to show that *per0* mutants lack the normal fluctuation in baseline calcium that are present in control flies, and furthermore that *per0* mutants do not show time-of-day differences in light responsiveness. This is entirely in line with the hypothesis that it is not actually the light response that is changing throughout the day, but instead the baseline calcium values. Their measure of light responsiveness is $\Delta F/F$, which is highly influenced by baseline values, which fluctuate across the day. In many experiments (those after Figure 2), baseline calcium values are not given, and graphs only show $\Delta F/F$ values. Thus, it is unclear whether there is actually reduced responsiveness at daytime time points, or if the cells are just starting at a higher baseline.

It is also not clear to me based on the authors' discussion of their results which possibility they favor, and I don't think that their experiments have successfully differentiated between the two. This applies even to subsequent experiments, for example those that show that temperature or DN1a ablation affects sLNv light responses. These could also be explained by alterations in basal calcium levels.

To help differentiate between these possibilities, it would be useful to quantify maximum fluorescence signal at the different time points, and show raw calcium data in addition to $\Delta F/F$ values. It may also be necessary to do electrophysiology or use a voltage indicator to directly assess neuronal activation in response to light. The bottom line is that this uncertainty must be addressed. If the major phenotype described in the paper simply reflects an artifact of how responsiveness is calculated, and derives from altered baseline calcium rather than *de facto* differences in temperature responsiveness across the day, then my enthusiasm for the manuscript would be lessened.

Minor concerns

1. The authors should explain why green light was used in Figure 1.
2. A few times, the authors claim that their calcium imaging demonstrates afterhyperpolarization, but this cannot be determined by looking at calcium signals. They would have to perform electrophysiological recordings (or use a genetically encoded voltage indicator) to see whether membrane potential is actually hyperpolarizing.
3. The authors conduct recordings in cry mutant flies to assess the contribution of cry to the sLNv light responses. Seeing no differences between the mutants and controls, they conclude that "These findings suggest that CRY is not essential for the circadian light response of s-LNvs." It should be clarified that this applies to short, acute light pulses. It is not clear whether CRY may contribute in the face of longer light pulses (which are more similar to what the fly would be exposed to in natural conditions).
4. Data for the effect of temperature on light-induced behavioral phase shifts is spread across Figure 7 and Figure EV2. It would be better to show all 3 temperatures on one graph, at least for one of either the phase delay or advance, so that the data can be easily compared (for example, all phase delay data could be shown in Figure 7 and then all phase advance in Figure EV2). I would also recommend showing a larger version of the first day of light shift compared across the different temperatures. The phenotype in control flies is slight and hard to see based on the way the data are currently depicted.
5. For calcium imaging $\Delta F/F$ heat maps (eg Figure 2C), do the individual lines at each time point correspond to different experiments?
6. Details of statistical tests are unclear: "Pairwise comparison between all groups was performed by one-way ANOVA followed by Tukey's test. Comparison between selected groups was performed by two-tailed unpaired Student's t-test, with the Bonferroni correction to P value for multiple comparisons." If the Tukey test is being done following ANOVA, this would allow for comparisons between all groups. It's not clear why a separate t-test is being run. Also, when graphs show asterisks (eg Figure 7B-E), it is unclear what they indicate (ie is it a comparison between specific groups? Which group(s) are statistically different?)
7. The description of how behavioral phase shift was calculated should include more information. The methods state: "To calculate the magnitude of the phase shift, daily activity profiles of individual flies were plotted, and the phase of the evening peak of each fly was determined manually, as described previously (Rieger et al., 2003)." Rieger et al. 2003 does not appear in the reference list. The authors should provide more detail about how they measured phase shifts instead of saying "as described previously"
8. "For the two-photon calcium imaging of live flies that experienced cooling and heating, the fly was submerged in saline". What part of the fly was submerged? Was the temperature being applied to just part of the fly?
9. What is the genetic background of the flies used in experiments? Were lines outcrossed to a common genetic background?

Referee #2:

This is a very elegant piece of work that reveals how temperature fine tunes the light responses of the circadian pacemaker neurons in *Drosophila*. I liked the paper very much - it's well-written and very clear.

I have a few comments

Nowhere in the title or the Abstract is 'Drosophila' mentioned. A curious reader would have to get to the bottom of p2 to find out that the study was focused on flies!

Methods. Am I right in thinking that different flies were used for the ZT measurements but the same fly for the different colors? Also, during ZT12-24, if there is a red light on, does that not interact with the blue/green/yellow stimuli? What were the wavelengths of the different lights and bandwidths?

Fig 2E Can the authors perform some sort of statistical analysis? SEMs are OK, they do not easily reveal what is significantly different to what - 95% confidence limits would be better. However, ANOVA plus post hoc tests would be much more informative. For example, a 2-way ANOVA of blue v green v yellow ON, then another ANOVA with OFF would provide a comprehensive analysis (repeated measure design if appropriate-see my comment above)

'We then examined whether l-LNvs are an alternative light input pathway to s-LNvs. l-LNvs are light-responsive neurons communicating with s-LNvs through Pigment Dispersing Factor (PDF) and PDF receptor (PDFR) signaling'. Reference required 'In the natural environment, ambient light and temperature serve as important coupled zeitgebers. While these environmental cues typically act in synchrony to influence circadian rhythms, their interactive effects on key circadian neurons remain unclear-specifically, whether light and temperature act synergistically or antagonistically.' Actually, this is not true. In the natural environment temperature cycles phase lag the light cycle by several hours (eg see Vanin et al 2012 Science).

Discussion. Some of the discussion was a bit confusing and the evolutionary/adaptive arguments were rather weak

'For instance, during colder seasons when light resources are scarce, shorter days and reduced light intensity at higher latitudes necessitate heightened light sensitivity in circadian pacemakers (Hidalgo et al, 2023; Ray et al, 2020).' Under shorter colder days with short photoperiods, Drosophila are in diapause and largely immobile, so the clock's' light sensitivity would not be very relevant. In contrast, reduced light sensitivity during hotter summer months seems more important because very long summer days disrupt circadian behaviour, particularly at higher latitudes eg > 55oN. Evolutionary relevant studies from, the Stanewsky and Costa groups also suggest that the reduction of light sensitivity at higher, colder latitudes is adaptive. Consequently, the authors might either tone down their 'just so' adaptive explanations or expand that section more comprehensively.

Given the focus on the glutamate expressing DN1a neurons, the authors might also wish to discuss the papers that reveal that DN1s also contribute to visual sensitivity. There are a number of specific papers on this. Consequently, the DN1's can potentially integrate the temperature/light signals before they reach the sLNvs.

Referee #3:

This study investigates how temperature modulates light-induced circadian phase shifts in *Drosophila melanogaster*. The use of in vivo two-photon calcium imaging under physiological conditions provides high-resolution, real-time data on circadian neuron activity, overcoming limitations of ex vivo approaches. The integration of a Peltier-based temperature modulation system with calcium imaging is technically impressive, enabling precise dissection of temperature-light interactions. The integration of these cutting-edge techniques and behavioral assays provides a compelling framework for understanding multisensory circadian regulation: they demonstrated that DN1a neurons inhibit s-LNvs via glutamatergic signaling, thereby calibrating light-induced PDF release and phase shifts, and they further showed that lower temperatures accelerate and higher temperatures slow phase shifts, aligning physiological and ecological relevance (e.g., seasonal adaptation). In summary, this study bridges a critical gap by elucidating how light and temperature-two dominant zeitgebers-interact at the neural circuit level, offering a model for multisensory integration in circadian systems. I have a couple of minor concerns of the manuscript before I could recommend its publication in the EMBO Journal.

1. The authors have comprehensively characterized the light-evoked responses of PDF fibers across different circadian timepoints in male *Drosophila*. Given the result evidence from studies suggesting that sLNv neurons and PDF neuropeptide signaling may exhibit sexual dimorphism (<https://pubmed.ncbi.nlm.nih.gov/38352594/>), it would significantly strengthen the study to examine whether female flies display comparable light response profiles under identical experimental conditions.

2. While the enhanced photic responses of sLNv neurons at ZT18 under low temperature conditions represent an intriguing finding, the absence of similar enhancement at ZT18 in both DN1a-ablated and DN1a>VGLUT RNAi experimental groups merits deeper discussion. Could the authors elaborate on potential mechanisms underlying this phenotypic divergence in the discussion?

3. Given that DN1p neurons serve as an additional source of inhibitory glutamate to sLNv and have been shown to closely monitor ambient temperature fluctuations (<https://www.nature.com/articles/nature25740>), it would be methodologically informative to employ tetanus toxin (TNT)-mediated synaptic blockade to specifically inhibit DN1p output. Subsequent assessment of sLNv photic responses under these experimental conditions could help disentangle the relative contributions of DN1a vs. DN1p circuits to temperature-dependent modulation of light responses.

4. All heat maps depicting calcium activity patterns should be supplemented with explicit time scale bars or time axis labels, which is critical for interpreting the temporal dynamics of neuronal responses.

5. The heat map presentation in Figure 6D would benefit from replacement with data obtained from flies with RNAi-mediated knockdown of glutamate transmission in DN1a neurons. This would more effectively demonstrate the specific effects of

glutamatergic manipulation compared to the current generalised presentation.

6. All figure legends must explicitly state the specific statistical methods used (e.g. paired t-test, one-way ANOVA with Tukey's post-hoc).

We extend our sincere gratitude to the three reviewers for their invaluable comments and constructive suggestions. In our revised submission, we have taken meticulous care to address each point raised by the reviewers, with the specific alterations to the manuscript thoughtfully highlighted in color. These revisions have undoubtedly enhanced the clarity and quality of the paper. We are delighted to observe that all three reviewers share our enthusiasm for the research findings presented in our manuscript. We firmly believe that our work offers novel perspectives and makes a valuable contribution to the existing literature.

Referee #1:

This manuscript from Tian et al. assesses the interaction between light and temperature in circadian clock entrainment. Much of the data are derived from 2-photon imaging of GCaMP signals in circadian clock neurons. Although previous studies have independently assessed light- or temperature-evoked responses in clock neurons, this manuscript is unique in that it investigates the interaction between these two parameters; namely, how temperature affects clock neuron responses to acute light pulses. The authors first suggest that light responsiveness of the sLNv clock neurons varies across the day, with larger responses during the dark period in LD conditions (or subjective night in DD conditions). As the sLNv neurons have previously been implicated in generating behavioral rhythms and in mediating light entrainment, these data offer a physiological correlate of these functional attributes. They then assess clock cell responsiveness to light pulses at different ambient temperatures and argue that light responses are enhanced at lower temperatures. They attribute the temperature modulation of sLNv light responses to inputs from temperature-sensitive DN1 clock cells, as DN1 ablation potentiates sLNv responses in a manner that mimics the effect of low temperature. Finally, they perform behavioral studies to show that reentrainment to phase shifts in light cycles is modified by temperature. At lower temperatures, there is a slight acceleration of behavioral re-entrainment.

Overall this is a very interesting paper that applies novel methods to address an important question in the field. However, there are a few major issues that need to be addressed, as detailed below.

We are grateful for the invaluable comments offered by the reviewer. As outlined below, we have diligently revised the manuscript to enhance its precision and informativeness. We firmly believe that these revisions hold significant importance in refining the quality of the manuscript.

Major Concerns

1. There are necessary details lacking from the experimental methods section that makes it difficult to adequately assess the merits of the manuscript.

-for the calcium imaging, what were the light conditions during the imaging? Were lights on for ZT0-12 time points? This is important to clarify because light conditions during imaging could alter baseline neuron firing and because ongoing ambient light could lead to activation light-response mechanisms, thereby affecting the acute response to the light pulse.

Thank you for the reviewer's thoughtful comments. In response, we have included detailed

experimental conditions in the Methods section:

"For experiments during the ZT0-12 time points, flies were removed from the LD entrainment incubator 30 minutes prior to imaging, and surgery to create a cuticle window was performed under normal lighting conditions. Flies were then placed under the two-photon microscope and allowed to acclimate for at least 5 minutes before the start of the experiment. It's important to note that two-photon microscopy imaging requires protection from light, so the environment during the actual imaging period was effectively dark at all time points except for the light pulse period.

Specifically, for the ZT12 time point, flies were removed at ZT11.5, and surgery was performed under normal light. At the ZT12 light-off time point, flies were transferred to the two-photon microscope for imaging.

For experiments during ZT12-24, flies were removed from the incubator under dark conditions and all surgical procedures were performed under dim red light to avoid circadian disruption. After surgery, the flies were imaged under the two-photon microscope."

-in several experiments, the authors have used the Clk856-GAL4 line to drive GCaMP expression in most clock neurons. They have also conducted the majority of their calcium imaging analysis on the dorsal projections of the clock neurons rather than the cell bodies. It is therefore unclear how they are able to differentiate between the different clock cell populations for their quantification (eg in Figure 1 C-E). This needs to be thoroughly explained and validated, because the authors claim that light responses are predominantly occurring in the sLNvs compared to other clock neurons. It would seem nearly impossible to differentiate the different individual clock cell populations when imaging from the densely intermingled dorsal projections.

We thank the reviewer for raising this important methodological issue. We have now significantly improved our explanation of how we differentiated between clock neuron populations in our imaging experiments and added it to the Methods section:

" For flies expressing GCaMP driven by CLK856-GAL4, we performed volumetric two-photon microscopy imaging focused on the dorsal brain region. Importantly, our analysis protocol allowed us to reliably distinguish between different clock neuron populations through careful 3D reconstruction of the brain using FIJI.

The cell bodies of DN1a, DN1p, and DN2 neurons were identified based on their distinct locations within the Z-stack, while s-LNv and LNd neurons were identified based on their characteristic fiber pattern. This approach takes advantage of several anatomical features that allow reliable differentiation:

DN1a neurons (two per hemisphere) have cell bodies positioned well above the dorsal fibers and separated from DN1ps.

DN1p neurons (several per hemisphere) are clustered near the dorsal fibers, either overlapping or slightly above them.

DN2 neurons (two per hemisphere) have cell bodies close to, but below, the dorsal fibers."

For clarity, we have added corresponding arrows to Figure 1C-D. It should be noted that Figure 1C-D show only representative neurons corresponding to the heat maps. For statistical analysis, we included all identifiable cell bodies and fibers for each neuron type in each brain, not just individual ROIs.

-for experiments in ILNvs (Figure 4A-B), what part of the ILNvs is being imaged? These cells do not project to the dorsal brain so it can't be the same method as used for the other cell types.

Thank you for this important comment. For the experiments imaging I-LNvs (Figure 4A-B), we positioned the fly head at a steeper angle and created an imaging window between the ocelli and the antennae to better visualize the I-LNv cell bodies. The imaging was performed at the anterior region of the fruit fly brain, specifically recording the cell bodies of I-LNvs rather than their projections. We have added a heat map showing the response of I-LNv cell bodies to light stimulation in the supplementary materials to better illustrate this approach.

I-LNv > *uas-GCaMP7*

2. In the text, the authors say that they used Dvpdf-GAL4, "a driver that can specifically label the dorsal projection fiber of sLNvs". What is the evidence that Dvpdf-GAL4 specifically labels the sLNv projections? The line has previously been described by the same authors as being expressed in both sLNv and LNd cells. The description in the text here is misleading, as they cite Figure 2A in support of the specificity of the driver. Figure 2A is a reconstruction of the sLNv cells from the FlyWire dataset. It would be better to show imaging of the actual Dvpdf-GAL4 expression pattern. Why not use the sLNv-specific split-GAL4 line (5500681) that they use in Figure EV1, as this is a cleaner driver that is specific for sLNvs? Furthermore, in Figure 2, it is labeled as Dvpdf-LexA. Could the authors clarify if they have used the GAL4 or LexA line?

Thanks for your helpful comments. We appreciate your pointing out the inconsistency in our description of the driver line. In fact, we used the Dvpdf-LexA driver in Figure 2, not Dvpdf-GAL4 as mistakenly stated in the text.

To clarify, our intent was to convey that the dorsal projection fiber of s-LNvs can be effectively labeled by the Dvpdf-LexA driver, not to claim that this line "specifically" labels only the dorsal fiber of s-LNvs. We have revised this statement in the text to read: "We further characterized the light responses of s-LNvs using Dvpdf-LexA, which also labels the dorsal projection fiber of s-LNvs (Figure 2A)".

In Figure 2A, we have included the expression pattern of Dvpdf-LexA and its co-labeling with s-LNvs labeled with PDF-GAL4 to demonstrate the labeling efficacy.

Regarding your suggestion to use the sLNv-specific split-GAL4 line (ss00681), we have performed additional experiments comparing light responses in s-LNvs labeled by three different drivers: Dvpdf-LexA, the sLNv-specific split-GAL4 line (ss00681), and PDF-GAL4. Our results confirm that s-LNvs labeled by all three lines show consistent circadian variation in response to light stimulation. The only notable difference was that GCaMP7s in sLNv showed a higher amplitude in response to light pulses at night compared to GCaMP6.

For our subsequent experiments involving multiple genetic crosses, we chose to use Dvpdf-LexA to maintain consistent genetic backgrounds across experimental strains, which is critical for the reliability of our comparative analyses.

3. I am confused about the authors' interpretation of the major findings of the paper. In the abstract and throughout the text, they say that higher temperatures suppress and lower temperatures enhance the responsiveness of s-LNV neurons to light. However, later in the results, they say "We hypothesized that the circadian rhythmicity in s-LNVs light responses might arise from the endogenous circadian clock modulating basal calcium levels across day and night. This predicts that higher basal calcium levels during daytime could limit light-induced activation through a ceiling effect, while lower nighttime levels could permit larger light-induced calcium increases." They then go on to show that *per0* mutants lack the normal fluctuation in baseline calcium that are present in control flies, and furthermore that *per0* mutants do not show time-of-day differences in light responsiveness. This is entirely in line with the hypothesis that it is not actually the light response that is changing throughout the day, but instead the baseline calcium values. Their measure of light responsiveness is $\Delta F/F$, which is highly influenced by baseline values, which fluctuate across the day. In many experiments (those after Figure 2), baseline calcium values are not given, and graphs only show $\Delta F/F$ values. Thus, it is unclear whether there is actually reduced responsiveness at daytime time points, or if the cells are just starting at a higher baseline.

It is also not clear to me based on the authors' discussion of their results which possibility they favor, and I don't think that their experiments have successfully differentiated between the two. This applies even to subsequent experiments, for example those that show that temperature or DN1a ablation affects s-LNV light responses. These could also be explained by alterations in

basal calcium levels.

To help differentiate between these possibilities, it would be useful to quantify maximum fluorescence signal at the different time points, and show raw calcium data in addition to $\Delta F/F$ values. It may also be necessary to do electrophysiology or use a voltage indicator to directly assess neuronal activation in response to light. The bottom line is that this uncertainty must be addressed. If the major phenotype described in the paper simply reflects an artifact of how responsiveness is calculated, and derives from altered baseline calcium rather than de facto differences in temperature responsiveness across the day, then my enthusiasm for the manuscript would be lessened.

Thank you for your thoughtful comments. We fully acknowledge your concern that our data representation using $\Delta F/F$ might not adequately differentiate between two possibilities: (1) actual changes in light responsiveness or (2) artifacts due to fluctuating baseline calcium levels. To address this important point, we conducted additional analyses of the basal calcium fluorescence values (F_{baseline}) in s-LNv fibers before light stimulation and the maximum fluorescence values (F_{max}) during light stimulation across temperature changes at different circadian timepoints (ZT6, ZT18).

Our new analysis reveals that for each fly tested at a specific timepoint, there was no significant difference in the basal fluorescence values before and after temperature changes. Importantly, the maximum fluorescence value (F_{max}) during light stimulation at lower temperatures was significantly higher than at normal temperature. Conversely, F_{max} at higher temperatures was significantly lower than at normal temperature. These results demonstrate that it is indeed the light response amplitude (as measured by F_{max}) that changes with temperature throughout the day, rather than merely reflecting differences in baseline calcium levels.

We applied the same analytical approach to our DN1a-inhibition experiments, calculating both F_{baseline} and F_{max} . Consistent with our temperature findings, DN1a-inhibited flies exhibited increased maximum GCaMP signal in response to light at most ZT timepoints, while baseline

values remained comparable to controls.

We have incorporated these new analyses as supplemental figures to provide clear evidence distinguishing between changes in baseline calcium and genuine alterations in light responsiveness.

Minor concerns

1. The authors should explain why green light was used in Figure 1.

We appreciate the reviewer's question about our choice of green light in Figure 1. Green light was chosen based on several important considerations:

Among the common colors of light in everyday life (red, yellow, green, and blue), green is an optimal choice for *Drosophila* studies. Blue light, which is close to UV wavelengths, can have harmful effects on flies. *Drosophila* have limited sensitivity to red light, making it suboptimal for our experiments. Previous literature (Lazopulo *et al*, 2019) has established that "flies have a green preference which requires rhodopsin-based visual photoreceptors and is controlled by the circadian clock," providing a scientific rationale for our choice.

While green light was shown in Figure 1, we also examined responses to other colors of light in subsequent experiments. These additional tests revealed that circadian neurons in *Drosophila* do not show high selectivity in their responses to different colors of light.

2. A few times, the authors claim that their calcium imaging demonstrates after hyperpolarization, but this cannot be determined by looking at calcium signals. They would have to perform electrophysiological recordings (or use a genetically encoded voltage indicator) to see whether membrane potential is actually hyperpolarizing.

Thank you for this important comment. We agree that calcium imaging cannot directly demonstrate after-hyperpolarization. We have revised our language throughout the manuscript to remove references to "after-hyperpolarization" and have instead described our observations in terms of calcium signaling dynamics without making claims about membrane potential.

3. The authors conduct recordings in cry mutant flies to assess the contribution of cry to the sLNv light responses. Seeing no differences between the mutants and controls, they conclude that "These findings suggest that CRY is not essential for the circadian light response of s-LNvs." It should be clarified that this applies to short, acute light pulses. It is not clear whether CRY may contribute in the face of longer light pulses (which are more similar to what the fly would be exposed to in natural conditions).

We appreciate the reviewer's insightful comment regarding the interpretation of our CRY mutant data. As suggested, we have clarified that our conclusion specifically applies to short, acute light pulses rather than making a broader statement about CRY's role in circadian light responses under all conditions.

We have revised our conclusion from "These findings suggest that CRY is not essential for the circadian light response of s-LNvs" to "These findings suggest that CRY is not essential for the circadian response of s-LNvs to short, acute light pulses."

4. Data for the effect of temperature on light-induced behavioral phase shifts is spread across Figure 7 and Figure EV2. It would be better to show all 3 temperatures on one graph, at least for one of either the phase delay or advance, so that the data can be easily compared (for example, all phase delay data could be shown in Figure 7 and then all phase advance in Figure EV2). I would also recommend showing a larger version of the first day of light shift compared across the different temperatures. The phenotype in control flies is slight and hard to see based on the way the data are currently depicted.

We have consolidated all phase delay data in Figure 7 and all phase advance data in Figure EV2, allowing for direct comparison across the three temperature conditions (as suggested). The behavioral data has been completely restructured according to the reviewer's comments, as shown in our revised figures.

Figure 7

Figure EV5

5. For calcium imaging $\Delta F/F$ heat maps (eg Figure 2C), do the individual lines at each time point correspond to different experiments?

We appreciate the reviewer's question regarding the interpretation of our calcium imaging $\Delta F/F$ heat maps. To clarify:

In these heat maps (such as Figure 2C), each row represents a region of interest (ROI). For each brain sample analyzed, we typically defined two ROIs—one from the left hemisphere and one from the right hemisphere. Each brain preparation corresponds to one independent experiment.

This experimental design allows us to capture bilateral responses while maintaining the identity of individual experimental replicates in our heat map visualizations.

We have now added this clarification in the method section.

6. Details of statistical tests are unclear: "Pairwise comparison between all groups was performed by one-way ANOVA followed by Tukey's test. Comparison between selected groups was performed by two-tailed unpaired Student's t-test, with the Bonferroni correction to P value for multiple comparisons." If the Tukey test is being done following ANOVA, this would allow for comparisons between all groups. It's not clear why a separate t-test is being run. Also, when graphs show asterisks (eg Figure 7B-E), it is unclear what they indicate (ie is it a comparison between specific groups? Which group(s) are statistically different?)

Thank you for your insightful comments regarding our statistical methodology. We recognize the confusion in our description. To clarify, the statistical methods listed in the methods section represent our complete analytical toolkit, not multiple tests applied to the same dataset. Each experiment utilized a specific statistical approach based on its design.

For clarity, we have now specified the exact statistical test used for each experiment in the corresponding figure legends, including what the asterisks represent (specific group comparisons and significance levels).

7. The description of how behavioral phase shift was calculated should include more information. The methods state: "To calculate the magnitude of the phase shift, daily activity profiles of individual flies were plotted, and the phase of the evening peak of each fly was determined manually, as described previously (Rieger et al.,2003)." Rieger et al. 2003 does not appear in the reference list. The authors should provide more detail about how they measured phase shifts instead of saying "as described previously"

We sincerely appreciate the reviewer's comments and apologize for our oversight regarding the reference. To address this, we have replaced the citation with a more relevant original study and provided a more detailed description of our method: "Specifically, the raw data were smoothed with a digital low-pass filter, with a cutoff frequency set at 6 hours. This level of smoothing enhanced the clarity of the activity profile while preserving the distinction between the lights-off and evening peaks. The time of the evening peak was then determined by manually removing subpeaks using the mouse pointer in MATLAB."

8. "For the two-photon calcium imaging of live flies that experienced cooling and heating, the fly was submerged in saline". What part of the fly was submerged? Was the temperature being applied to just part of the fly?

Thank you for this important clarification request. The temperature control system used in our experiments is described in detail in Li et al (Li *et al*, 2024). To summarize, the fly brain was submerged in saline, while temperature modulation was achieved using a Peltier element attached to a metallic fly holder.

Because of the low thermal resistance between the saline solution and the Peltier element, temperature changes in the brain closely followed those applied via the Peltier device. The fly brain was in direct contact with the metallic holder, which ensured efficient heat transfer. As a

result, the temperature of the fruit fly body changed slightly later than that of the brain. However, this minor delay does not impact our results, as we are specifically recording neural activity in the fly brain, where temperature changes are directly relevant to our study.

9. What is the genetic background of the flies used in experiments? Were lines outcrossed to a common genetic background?

Most of the fly strains used in our study were obtained from the Bloomington Stock Center and are on the w¹¹¹⁸ genetic background. The exceptions are *norpA* and *Rh6-GAL4*, which are on the w^{**} background. No additional outcrossing to a common genetic background was performed

Referee #2:

This is a very elegant piece of work that reveals how temperature fine tunes the light responses of the circadian pacemaker neurons in *Drosophila*. I liked the paper very much - it's well-written and very clear.

I have a few comments

Nowhere in the title or the Abstract is '*Drosophila*' mentioned. A curious reader would have to get to the bottom of p2 to find out that the study was focused on flies!

Thank you very much for your valuable feedback. We apologize for not mentioning the model organism earlier, as this would have improved clarity for readers. In response to your suggestion, we have made the following changes:

Updated the Abstract to explicitly mention *Drosophila* earlier: "Here, we employed an in vivo two-photon calcium imaging and precise temperature control, to investigate how the circadian clock integrates light and temperature inputs in circadian neurons in *Drosophila melanogaster*."

Methods. Am I right in thinking that different flies were used for the ZT measurements but the same fly for the different colors? Also, during ZT12-24, if there is a red light on, does that not interact with the blue/green/yellow stimuli? What were the wavelengths of the different lights and bandwidths?

Thank you for your insightful comments. You are correct that different flies were used for the ZT measurements, while the same fly was used for testing responses to different colors.

Regarding ZT12-24, fruit flies are maintained in complete darkness except during brain dissection. During dissection, we use only a very faint red light which minimally impacts the experiment because: (1) the dissection time is brief (less than two minutes), (2) red light falls outside the fruit fly visual spectrum and cannot effectively degrade circadian proteins, and (3) we use weak red illumination. Additionally, before conducting two-photon imaging, flies are allowed to adapt to dark conditions for a minimum of five minutes.

The wavelengths of the light stimuli are precisely controlled: blue (460-462 nm), green (522-525 nm), and yellow (570-572 nm). We have now added these specific wavelength details to our methods section.

Fig 2E Can the authors perform some sort of statistical analysis? SEMs are OK, they do not easily reveal what is significantly different to what - 95% confidence limits would be better. However, ANOVA plus post hoc tests would be much more informative. For example, a 2-way ANOVA of blue v green v yellow ON, then another ANOVA with OFF would provide a comprehensive analysis (repeated measure design if appropriate-see my comment above)

Thank you for your constructive comments regarding Figure 2E. We apologize for the lack of statistical analysis in the original submission. As you suggested, we have now incorporated a comprehensive statistical approach and show the data with 95% confidence limits as below. For Figure 2E, we employed two separate two-way ANOVAs: one analyzing blue vs. green vs. yellow in the ON condition, and another for the OFF condition. We have also included post-hoc tests to identify specific significant differences between groups. The detailed results of these statistical analyses are provided in a supplementary table. Your recommendations have significantly improved the statistical rigor of our work.

Table S1

No	Tukey's multiple comparisons test	Predicted (LS) mean diff.	95.00% CI of diff.	Significant?	Summary	Adjusted P Value
1	Blue:ZT1 vs. Blue:ZT6	0.1169	-0.3795 to 0.6133	No	ns	0.9997
2	Blue:ZT1 vs. Blue:ZT12	-0.2006	-0.7147 to 0.3136	No	ns	0.9784
3	Blue:ZT1 vs. Blue:ZT18	-0.7636	-1.249 to -0.2779	Yes	****	<0.0001
4	Blue:ZT1 vs. Green :ZT1	0.01472	-0.1312 to 0.1606	No	ns	>0.9999
5	Blue:ZT1 vs. Yellow:ZT1	-0.008707	-0.1546 to 0.1372	No	ns	>0.9999
6	Blue:ZT6 vs. Blue:ZT12	-0.3174	-0.8189 to 0.1840	No	ns	0.6213
7	Blue:ZT6 vs. Blue:ZT18	-0.8805	-1.353 to -0.4084	Yes	****	<0.0001
8	Blue:ZT6 vs. Green :ZT6	-0.0207	-0.1593 to 0.1179	No	ns	>0.9999
9	Blue:ZT6 vs. Yellow:ZT6	-0.08104	-0.2196 to 0.05753	No	ns	0.7301
10	Blue:ZT12 vs. Blue:ZT18	-0.563	-1.054 to -0.07216	Yes	*	0.0107
11	Blue:ZT12 vs. Green :ZT12	-0.03846	-0.1870 to 0.1101	No	ns	0.9994
12	Blue:ZT12 vs. Yellow:ZT12	-0.07936	-0.2279 to 0.06922	No	ns	0.8289
13	Blue:ZT18 vs. Green :ZT18	-0.03951	-0.1717 to 0.09270	No	ns	0.9977
14	Blue:ZT18 vs. Yellow:ZT18	-0.04847	-0.1807 to 0.08374	No	ns	0.9867
15	Green :ZT1 vs. Green :ZT6	0.08146	-0.4150 to 0.5779	No	ns	>0.9999
16	Green :ZT1 vs. Green :ZT12	-0.2537	-0.7678 to 0.2604	No	ns	0.8912
17	Green :ZT1 vs. Green :ZT18	-0.8178	-1.304 to -0.3321	Yes	****	<0.0001
18	Green :ZT1 vs. Yellow:ZT1	-0.02343	-0.1693 to 0.1225	No	ns	>0.9999
19	Green :ZT6 vs. Green :ZT12	-0.3352	-0.8366 to 0.1663	No	ns	0.5374
20	Green :ZT6 vs. Green :ZT18	-0.8993	-1.371 to -0.4272	Yes	****	<0.0001
21	Green :ZT6 vs. Yellow:ZT6	-0.06034	-0.1989 to 0.07823	No	ns	0.9518
22	Green :ZT12 vs. Green :ZT18	-0.5641	-1.055 to -0.07321	Yes	*	0.0104
23	Green :ZT12 vs. Yellow:ZT12	-0.0409	-0.1895 to 0.1077	No	ns	0.9989
24	Green :ZT18 vs. Yellow:ZT18	-0.008965	-0.1412 to 0.1232	No	ns	>0.9999
25	Yellow:ZT1 vs. Yellow:ZT6	0.04455	-0.4519 to 0.5410	No	ns	>0.9999
26	Yellow:ZT1 vs. Yellow:ZT12	-0.2712	-0.7853 to 0.2429	No	ns	0.8401
27	Yellow:ZT1 vs. Yellow:ZT18	-0.8033	-1.289 to -0.3176	Yes	****	<0.0001
28	Yellow:ZT6 vs. Yellow:ZT12	-0.3158	-0.8172 to 0.1857	No	ns	0.6291
29	Yellow:ZT6 vs. Yellow:ZT18	-0.8479	-1.320 to -0.3758	Yes	****	<0.0001
30	Yellow:ZT12 vs. Yellow:ZT18	-0.5321	-1.023 to -0.04127	Yes	*	0.0213

'We then examined whether 1-LNVs are an alternative light input pathway to s-LNVs. 1-LNVs are light-responsive neurons communicating with s-LNVs through Pigment Dispersing Factor (PDF) and PDF receptor (PDFR) signaling'. Reference required

Thank you for highlighting this oversight. We have added the appropriate references as suggested: "We then examined whether l-LNVs are an alternative light input pathway to s-LNVs. l-LNVs are light-responsive neurons communicating with s-LNVs through Pigment Dispersing Factor (PDF) and PDF receptor (PDFR) signaling (Shafer et al, 2008; Shang et al, 2008)."

'In the natural environment, ambient light and temperature serve as important coupled zeitgebers. While these environmental cues typically act in synchrony to influence circadian rhythms, their interactive effects on key circadian neurons remain unclear-specifically, whether light and temperature act synergistically or antagonistically. 'Actually, this is not true. In the natural environment temperature cycles phase lag the light cycle by several hours (eg see Vanin et al 2012 Science).

Thank you for this important correction. We appreciate your expertise regarding the natural relationship between light and temperature cycles. We have modified our statement to more accurately reflect the scientific evidence: 'In the natural environment, ambient light and temperature serve as important coupled zeitgebers. While environmental temperature cycles typically phase lag the light cycle by several hours (Vanin et al., 2012 Science), their interactive

effects on key circadian neurons remain unclear—specifically, whether light and temperature act synergistically or antagonistically.

Discussion. Some of the discussion was a bit confusing and the evolutionary/adaptive arguments were rather weak 'For instance, during colder seasons when light resources are scarce, shorter days and reduced light intensity at higher latitudes necessitate heightened light sensitivity in circadian pacemakers (Hidalgo et al, 2023; Ray et al, 2020).' Under shorter colder days with short photoperiods, *Drosophila* are in diapause and largely immobile, so the clock's 'light sensitivity would not be very relevant. In contrast, reduced light sensitivity during hotter summer months seems more important because very long summer days disrupt circadian behaviour, particularly at higher latitudes eg > 55°N. Evolutionary relevant studies from, the Stanewsky and Costa groups also suggest that the reduction of light sensitivity at higher, colder latitudes is adaptive. Consequently, the authors might either tone down their 'just so' adaptive explanations or expand that section more comprehensively.

We appreciate the reviewer's insightful critique of our evolutionary arguments. We acknowledge the limitations in our original discussion and have revised our approach accordingly. As the reviewer correctly points out, *Drosophila* typically enter diapause during shorter, colder days, rendering clock light sensitivity less relevant during these conditions.

In our revised discussion, we have toned down our assertions regarding increased light sensitivity during colder seasons and instead emphasized the adaptive significance of reduced light sensitivity during hotter summer months, particularly at higher latitudes (>55°N). At these latitudes, extended photoperiods can disrupt circadian behavior, making the modulation of the circadian clock's light sensitivity a potentially important mechanism for maintaining appropriate behavioral rhythms. This perspective aligns with the evolutionary studies from the Stanewsky and Costa groups that the reviewer helpfully referenced.

We thank the reviewer for guiding us toward a more scientifically accurate and comprehensive discussion of these adaptive mechanisms

Given the focus on the glutamate expressing DN1a neurons, the authors might also wish to discuss the papers that reveal that DN1s also contribute to visual sensitivity. There are a number of specific papers on this. Consequently, the DN1's can potentially integrate the temperature/light signals before they reach the sLNvs.

Thank you for highlighting the important connection between DN1 neurons and visual sensitivity. We appreciate this valuable suggestion and the reference to relevant literature in this area. In response, we conducted additional experiments specifically measuring DN1 responses to light pulses at different circadian time points. As illustrated in our new data, we used R18H11-LexA to drive GCaMP expression in DN1s (including both DN1as and DN1ps). Our results demonstrate that these DN1s did not exhibit significant responses to light pulses at either daytime (ZT6) or nighttime (ZT18). Considering these findings alongside previous literature, we propose that DN1s may integrate temperature/light signals through indirect pathways rather than through direct and acute responses to light.

Referee #3:

This study investigates how temperature modulates light-induced circadian phase shifts in *Drosophila melanogaster*. The use of in vivo two-photon calcium imaging under physiological conditions provides high-resolution, real-time data on circadian neuron activity, overcoming limitations of ex vivo approaches. The integration of a Peltier-based temperature modulation system with calcium imaging is technically impressive, enabling precise dissection of temperature-light interactions. The integration of these cutting-edge techniques and behavioral assays provides a compelling framework for understanding multisensory circadian regulation: they demonstrated that DN1a neurons inhibit s-LNvs via glutamatergic signaling, thereby calibrating light-induced PDF release and phase shifts, and they further showed that lower temperatures accelerate and higher temperatures slow phase shifts, aligning physiological and ecological relevance (e.g., seasonal adaptation). In summary, this study bridges a critical gap by elucidating how light and temperature—two dominant zeitgebers—interact at the neural circuit level, offering a model for multisensory integration in circadian systems. I have a couple of minor concerns of the manuscript before I could recommend its publication in the EMBO Journal.

1. The authors have comprehensively characterized the light-evoked responses of PDF fibers across different circadian timepoints in male *Drosophila*. Given the result evidence from studies suggesting that s-LNv neurons and PDF neuropeptide signaling may exhibit sexual dimorphism (<https://pubmed.ncbi.nlm.nih.gov/38352594/>), it would significantly strengthen the study to examine whether female flies display comparable light response profiles under identical experimental conditions.

Thank you for highlighting this important consideration regarding potential sexual dimorphism in neuronal responses. In response to your suggestion, we conducted additional experiments examining light-evoked responses in female *Drosophila* under identical experimental conditions. As shown in our new data, s-LNv neurons in female flies exhibit the same pattern of circadian modulation as males, with lower amplitude responses during daytime (ZT6) and higher amplitude responses during nighttime (ZT18). These findings indicate that the circadian regulation of light sensitivity in s-LNv neurons is consistent between sexes, suggesting this

fundamental property of the circadian circuit is not sexually dimorphic. This new evidence strengthens our overall conclusions about the temporal gating of light responses in the *Drosophila* circadian system

2. While the enhanced photic responses of sLNv neurons at ZT18 under low temperature conditions represent an intriguing finding, the absence of similar enhancement at ZT18 in both DN1a-ablated and DN1a>VGLUT RNAi experimental groups merits deeper discussion. Could the authors elaborate on potential mechanisms underlying this phenotypic divergence in the discussion?

Thank you for this insightful observation regarding the differential temperature effects at ZT18. We have expanded our discussion to address this important phenotypic divergence.

The interaction between ambient light and temperature in regulating circadian rhythms represents a crucial yet understudied research area. Our findings reveal a compensatory mechanism whereby key circadian pacemakers calibrate their light-induced firing patterns and PDF peptide release according to temperature variations—with high temperatures reducing light-induced pacemaker firing and low temperatures enhancing light responses.

Interestingly, when mimicking a cold environment through DN1a ablation or DN1a>VGLUT RNAi, the enhanced response to light pulses was observed specifically during daytime but not at ZT18, unlike the pattern seen in wild-type flies under actual low temperature conditions. We propose several potential mechanisms to explain this discrepancy:

First, the apparent lack of enhancement at ZT18 may reflect a ceiling effect of the GCaMP calcium indicator, which could mask further increases in calcium signaling during nighttime when responses are already elevated. Alternatively, this phenomenon may relate to the intrinsically low calcium levels in DN1a neurons at night (Li et al., 2024), diminishing the impact of DN1a ablation during this time window. Under this scenario, removing an already minimally active inhibitory input would produce limited effects compared to its removal during daytime when DN1a activity is higher.

These findings highlight the complex temporal dynamics of temperature-light integration in the circadian network and suggest that additional regulatory mechanisms may operate specifically during nighttime. Further investigations will be necessary to fully elucidate the intricate mechanisms by which these environmental zeitgebers interact to modulate circadian function across the 24-hour cycle.

3. Given that DN1p neurons serve as an additional source of inhibitory glutamate to s-LNv and have been shown to closely monitor ambient temperature fluctuations (<https://www.nature.com/articles/nature25740>), it would be methodologically informative to employ tetanus toxin (TND-mediated synaptic blockade to specifically inhibit DN1p output. Subsequent assessment of s-LNv photic responses under these experimental conditions could help disentangle the relative contributions of DN1a vs. DN1p circuits to temperature-dependent modulation of light responses.

Thank you for your insightful suggestion to employ tetanus toxin (TNT)-mediated synaptic blockade to differentiate between DN1a and DN1p circuit contributions. We appreciate this methodological recommendation and have implemented it in our study.

Following your guidance, we conducted additional experiments examining s-LNv light responses in flies with specifically inhibited DN1p output at both daytime (ZT6) and nighttime (ZT18). Our results reveal that while s-LNvs maintain their characteristic temporal gating pattern—exhibiting lower amplitude responses during daytime and higher amplitude responses at night—the overall magnitude of these responses was attenuated at both timepoints when DN1p output was blocked.

Notably, these results align with previous findings that DN1p neurons are inhibited by higher temperatures (Yadlapalli et al., Nature, 2018). Thus, our observed attenuation of s-LNv light responses following DN1p output blockade effectively mimics a high-temperature environment. These new data provide additional mechanistic insight into the temperature-mediated regulation of circadian photosensitivity and further strengthen our conclusion that temperature serves as a critical modulatory factor fine-tuning s-LNv responses to light across the circadian cycle.

4. All heat maps depicting calcium activity patterns should be supplemented with explicit time scale bars or time axis labels, which is critical for interpreting the temporal dynamics of neuronal responses.

Thank you for highlighting this important methodological consideration. We have incorporated explicit time scale bars on all heat maps depicting calcium activity patterns, including those in Figures 1C-D, 2C, 4C, 4E, 5A, and 5D. This addition enhances the interpretability of the temporal dynamics in our neuronal response data.

5. The heat map presentation in Figure 6D would benefit from replacement with data obtained from flies with RNAi-mediated knockdown of glutamate transmission in DN1a neurons. This would more effectively demonstrate the specific effects of glutamatergic manipulation compared to the current generalised presentation.

We appreciate your suggestion to strengthen the mechanistic specificity of our findings. As recommended, we have replaced the heat map in Figure 6D with new data obtained from flies with RNAi-mediated knockdown of glutamate transmission specifically in DN1a neurons. This revision more effectively demonstrates the targeted effects of glutamatergic manipulation in the DN1a population compared to our previous generalized presentation.

6. All figure legends must explicitly state the specific statistical methods used (e.g. paired t-test, one-way ANOVA with Tukey's post-hoc).

Thank you for this important reminder regarding statistical reporting standards. We have thoroughly revised all figure legends to explicitly state the specific statistical methods employed for each analysis. This includes detailed information about the test type (e.g., paired t-test, one-way ANOVA, two-way ANOVA) and, where applicable, the post-hoc comparison methods used (e.g., Tukey's post-hoc).

Lazopulo S, Lazopulo A, Baker JD, Syed S (2019) Daytime colour preference in depends on the circadian clock and TRP channels. *Nature* 574: 108-111

Li H, Li Z, Yuan X, Tian Y, Ye W, Zeng P, Li XM, Guo F (2024) Dynamic encoding of temperature in the central circadian circuit coordinates physiological activities. *Nat Commun* 15: 2834

Dear Fang,

Thank you again for submitting your revised manuscript (EMBOJ-2024-120009R) to The EMBO Journal for our consideration, and for your patience during peer review. As I have already informed you, your manuscript has been seen by the three original referees who had previously assessed the first version of your manuscript, and we have received their comments, which you can find below.

As you will see, all three referees recognize that the manuscript has been significantly improved during revision, and that many of the initially raised concerns have been adequately addressed. Referee #3 has no further comments and recommends the publication of the manuscript, but referees #1 and #2 still identify several limitations and point out some concerns that have not been fully addressed. We have discussed their comments extensively and largely agree with them that the points they make are constructive and reasonable, calling for either additional experimental work or textual revision and clarification to be fully addressed in another version of the manuscript. In light of this input and our discussions, I would like to invite you to submit a final version of your manuscript fully addressing all remaining concerns. Please include in your resubmission a point-by-point letter with detailed and complete responses to all comments, also describing and explaining any changes to the manuscript.

From the editorial side, there are also a number of changes and corrections we need you to make in the next version of your manuscript, before we can proceed further with its handling:

- Please include a list of up to 5 relevant keywords to enhance the search engine discoverability of the manuscript after the Abstract of your revised manuscript.
- If you have directly deposited your Source Data to the BioStudies database, as you mention in your "Data availability" statement, please include in this statement the dataset identifier and the specific URL (link) to the deposited data.
- The author contributions statement should be removed from the manuscript file. Instead, we use CRediT to specify the contributions of each author in the journal submission system. Please feel free to use the free text box to provide more detailed descriptions during submission. See also our guide to authors for more information:
<https://www.embopress.org/page/journal/14602075/authorguide#authorshipguidelines>.
- Figure panel callouts should be listed sequentially.
- Table S1 should be renamed to "Table EV1" and callouts for it should be provided where appropriate in the manuscript.
- We also noticed that callouts for Figure EV5 are also missing; please make sure that all Figure panels are called out (sequentially) in your revised manuscript.
- Please specify in the last column of your Author Checklist only the section(s) of the manuscript where the requested information can be found; the information per se should be provided in the manuscript, not in the Author Checklist.
- Materials and methods need to be described in the manuscript using our structured methods format, which is now required for all research articles. According to this format, the Methods section includes a single "Reagents and Tools Table" -listing key reagents, primers, experimental models, software and relevant equipment including their sources and relevant identifiers- followed by a "Methods and Protocols" section describing the methods. Please download and fill our Reagents and Tools Table template (.docx), which you can find in our author guide:
<https://www.embopress.org/page/journal/14602075/authorguide#structuredmethods>. When submitting your revised manuscript, please do not include the Reagents and Tools Table in the Methods section of the manuscript but instead upload it as a separate file choosing the file type "Reagent Table".
- Please note that EMBO press papers are accompanied online by:
 - A) a short (2 sentences) summary of the findings and their significance,
 - B) 2-5 short bullet points highlighting the key results, and
 - C) a synopsis image in .jpg or .png format that is exactly 550 pixels wide and 300-600 pixels high (the height is variable). Please note that the text needs to be legible at the final size.Please upload this information along with your revised manuscript (the text for A and B should be provided in a separate Word file).
- During our routine checks, our data editors have raised the following queries regarding Figures, data, and legends. Please make sure that the following requests are fully addressed in the next version of your manuscript:
 1. Please provide the exact p values in the legends of Figures 3B, C; 4D, F; 5B, C, F; 6C, E, F; 7B-E; EV1 A; EV3 E-H; EV4 B; EV5 B, D.
 2. Please note that information related to "n" is missing in the legends of Figures 6C, 7B-E, EV1A-C; EV4 A, B; EV5 B-E.

3. Please note that the error bars are not defined in the legends of Figures 7B-E.

- Table EV1 -which is currently uploaded as a .tiff file- should be in Word, Excel or PDF format, and its legend should be renamed to "Table EV1" (instead of "Table S1").

- Movie files as well as their corresponding callouts throughout the manuscript should be renamed to "Movie EV1-EV2", and their legends should be removed from the main manuscript file and instead zipped individually with each movie file.

- The order of the manuscript sections must be corrected as follows: Title page - Abstract and Keywords - Introduction - Results - Discussion - Methods - Data Availability - Acknowledgements - Disclosure and Competing Interests Statement - References - Figure Legends - main Tables (if there are any) - Expanded View Figure Legends.

Please also note that as part of the EMBO publications' Transparent Editorial Process, The EMBO Journal publishes online a Peer Review File along with each accepted manuscript. This File will be published in conjunction with your paper and will include the referee reports, your point-by-point response and all pertinent correspondence relating to the manuscript. You can opt out of this by letting the editorial office know (contact@embojournal.org). If you do opt out, the Peer Review File link will point to the following statement: "No Peer Review File is available with this article, as the authors have chosen not to make the review process public in this case."

We look forward to seeing a final version of your manuscript as soon as possible. Please let us know if you have any questions and use this link to submit your revision: <https://emboj.msubmit.net/cgi-bin/main.plex>.

Best wishes,

Ioannis

Referee #1:

The authors have conducted additional data collection and analysis to address several of my major concerns. Overall, I am convinced that temperature alters the acute response of s-LNV neurons to light, and that this is associated with a decrease in the magnitude of re-entrainment to phase advances and delays induced by changing in light regime. Furthermore, it is plausible that this arises in part due to input from DN1a neurons. However, a few issues remain unresolved, particularly regarding the claim that light responsiveness oscillates across the day:

Major Points:

1. The authors have now provided details on how they were able to differentiate between the different clock neuron subsets in their 2-photon imaging with Clock856-GAL4 driving expression of GCaMP. Specifically, they have identified DN1 neuron types based on anatomical location and s-LNV and LNd neurons based on their characteristic fiber projections. This new information raises several additional concerns. First, it may not be appropriate to directly compare calcium signals in cell bodies (for DN cells and I-LNVs) and in dorsal fibers (for s-LNV and LNd), especially as the authors note differences in baseline calcium oscillations in these two compartments. This does not invalidate the effects seen within individual populations, but it does make it difficult to determine whether the differences between these cell types could be due to differences in processing in the cell body vs projections. The authors should acknowledge this limitation.

Second, I can't tell from the arrowheads that have been added to Figure 1C and D what exactly is being measured for the LNd neurons. One of the gray arrowheads in 1C appears to be pointing off the image. I also am not convinced that the dorsal projection of the s-LNV neurons can be unambiguously distinguished from the central projection of the LNd and from the projections of all the other clock neurons. In this respect, the new data provided in the author rebuttal using the SS00681 line is important because it eliminates the confound of disambiguating the different clock cells. I would advocate for including this SS00681 data into the paper.

2. In my initial review, I raised a concern about the use of $\Delta F/F$ to determine light responsiveness. Because there are

considerable differences in baseline calcium before light presentation, the major effect they are seeing of differential light responsiveness in sLNv cells throughout the day could be simply due to a ceiling effect. The authors have provided additional analysis to attempt to address this point, but it seems as if they have misunderstood my concern. In the rebuttal letter they state:

"Thank you for your thoughtful comments. We fully acknowledge your concern that our data representation using $\Delta F/F$ might not adequately differentiate between two possibilities: (1) actual changes in light responsiveness or (2) artifacts due to fluctuating baseline calcium levels.

To address this important point, we conducted additional analyses of the basal calcium fluorescence values (F_{baseline}) in s-LNV fibers before light stimulation and the maximum fluorescence values (F_{max}) during light stimulation across temperature changes at different circadian timepoints (ZT6, ZT18)."

In an addition to the results section, the authors say "Importantly, F_{max} during light stimulation was significantly higher at lower temperatures compared to normal temperatures (Fig EV3E-F), while at higher temperatures, F_{max} was significantly lower than at normal temperatures (Fig EV3G-H). These findings indicate that the light response amplitude (as measured by F_{max}) changes with temperature throughout the day."

The data do convincingly show that changes in temperature alter the light responsiveness of the s-LNV neurons, but I don't see anything in the analysis that confirms that light response amplitude changes with temperature throughout the day. That would require a comparison of F_{max} at different time points. A similar analysis would be required to differentiate whether the main effect detailed in Figs 1-2 (that light responsiveness oscillates across the day) is due to changes in baseline calcium or instead to actual differences in light-evoked calcium increases. In fact, the new data provided in Fig. EV4 suggests that it is the former. Control flies have no differences in peak calcium after light stimulation but do have differences in baseline values. Can the authors show that F_{max} in response to light stimulus is oscillating across the day? If not, then it should be explicitly acknowledged in the text.

3. The authors have added more information to the methods regarding how phase shift was determined (for Figs 8 and EV5), but it is still not clear to me exactly what was done: "Specifically, the raw data were smoothed with a digital low-pass filter, with a cutoff frequency set at 6 hours. This level of smoothing enhanced the clarity of the activity profile while preserving the distinction between the lights-off and evening peaks. The time of the evening peak was then determined by manually removing subpeaks using the mouse pointer in MATLAB."

What does it mean to "manually remove subpeaks"? How were subpeaks and major peak identified? And what is actually used to determine phase? In Fig 7, the red lines looks like they are showing the phase of peak activity. In Fig EV5, they appear to mark the onset of the upwards slope of activity. And why in Fig 7 are some of these dashed red lines vertical while others are slanted (eg the dashed red line on D1 of the leftmost figure is vertical, while the subsequent days are slanted)? This is important to clarify because the manner through which phase was determined could influence the relatively small differences measured in day 1 phase shifts in control flies

Minor Points:

1. The authors should include the details of how they conducted the l-LNV calcium imaging in the methods section.
2. In the last section of the results (Temperature fine-tuned the light-induced phase shifts), it seems like many of the references to figures are incorrect as the authors have updated the figure layouts but did not update the figure references.
3. Could the authors explain why they have used multiple individual t-tests for statistical comparison in figs 4, 6 and 7 rather than ANOVA with Tukey test? And justify why in Fig 7 they have only compared between 24 and 29°C in Cry02 mutants, rather than making all comparisons (especially since they are arguing that there is a difference in phase shift on D1 for the control flies)?

Referee #2:

The ms is much improved and I commend the authors for replying constructively to my comments (and those of the other referees). I have a couple of minor outstanding issues.

1 Title 'Glutamate-Gated Light-Induced PDF Release in Circadian Pacemakers Modulates Temperature-Dependent Phase Shifts in *Drosophila*' - please add species

2 'reducing light-induced pacemaker firing during hot seasons and enhancing responses during colder seasons-to potentially optimize evolutionary fitness. the last bit about fitness is a stretch because there is no clear route from the phenotype to Darwinian fitness, so please cut that - it would make an evolutionary biologist wince!!!

2. The Discussion is rather unusual because it only cites 3 references!! A Discussion is supposed to integrate the authors' results with those from the rest of the literature, so this seems very one-dimensional. For example there is a literature on DN1s 'talking' to sLNvs (see Guo et al 2016; Murad et al 2007; Picot et al 2007; Stoleru et al 2007) and the role of glutamate in these interactions (Hamasaka et al 2005, 2007; Azevedo et al 2020) to name a few. The authors really need to do a better job to complement their findings with the literature and particularly for EMBO J.

Referee #3:

The authors have adequately addressed all my previous concerns and I recommend its publication in the EMBO J.

We sincerely appreciate the three reviewers for their constructive suggestions and insightful feedback. In our revised submission, we have meticulously addressed each of their comments and clearly marked all modifications in color for easy reference. These revisions have significantly improved the clarity and quality of the paper. We are delighted that all three reviewers share our enthusiasm for the research findings presented in our manuscript. We firmly believe that our work provides novel perspectives and makes a meaningful and valuable contribution to the existing research.

Referee #1:

The authors have conducted additional data collection and analysis to address several of my major concerns. Overall, I am convinced that temperature alters the acute response of sLNv neurons to light, and that this is associated with a decrease in the magnitude of re-entrainment to phase advances and delays induced by changing in light regime. Furthermore, it is plausible that this arises in part due to input from DN1a neurons. However, a few issues remain unresolved, particularly regarding the claim that light responsiveness oscillates across the day:

We sincerely thank the reviewer for the invaluable feedback. As detailed below, we have carefully addressed each comment to improve the manuscript's precision and clarity. We are confident that these revisions have significantly enhanced the overall quality of the work.

Major Points:

1. The authors have now provided details on how they were able to differentiate between the different clock neuron subsets in their 2-photon imaging with Clock856-GAL4 driving expression of GCaMP. Specifically, they have identified DN1 neuron types based on anatomical location and s-LNv and LNd neurons based on their characteristic fiber projections. This new information raises several additional concerns. First, it may not be appropriate to directly compare calcium signals in cell bodies (for DN cells and l-LNvs) and in dorsal fibers (for s-LNv and LNd), especially as the authors note differences in baseline calcium oscillations in these two compartments. This does not invalidate the effects seen within individual populations, but it does make it difficult to determine whether the differences between these cell types could be due to differences in processing in the cell body vs projections. The authors should acknowledge this limitation.

Thank you for the reviewer's insightful comment. We acknowledge the valid concern regarding the comparison of calcium signals between cell bodies (DN cells and l-LNvs) and dorsal fibers (s-LNv and LNd).

As the reviewer correctly points out, this methodological limitation does not invalidate the effects observed within individual populations, but it does introduce complexity when making direct comparisons between cell types. This limitation is evident in the increased variability of calcium signals in the dorsal fibers of LNd neurons, as shown in the right panel of Figure 1E.

To address this concern, we have added the following statement in the Method section: "In particular, due to the different projection patterns of clock neurons and the limitations of the cuticle window during two-photon imaging, we had to monitor different neuronal compartments

in different cell types. Some neurons can be identified and imaged at the cell body, while others, such as LNds, were only accessible at the processes. This compartmental difference is a limitation when comparing response patterns between cell bodies and neuronal projections, as the baseline calcium dynamics may differ between these compartments".

Second, I can't tell from the arrowheads that have been added to Figure 1C and D what exactly is being measured for the LNd neurons. One of the gray arrowheads in 1C appears to be pointing off the image. I also am not convinced that the dorsal projection of the s-LNv neurons can be unambiguously distinguished from the central projection of the LNds and from the projections of all the other clock neurons. In this respect, the new data provided in the author rebuttal using the SS00681 line is important because it eliminates the confound of disambiguating the different clock cells. I would advocate for including this SS00681 data into the paper.

Thank you for this meticulous observation. We appreciate your attention to detail regarding the identification of LNd neurons and s-LNv projections in Figure 1C and D.

To improve clarity, we have added corresponding arrows to the upper panel showing baseline images in Figure 1C and D. Additionally, we have replaced the representative panel in Figure 1D with a clearer image for LNds. While LNd fibers can be more easily identified in Z-stack images, they indeed become less distinguishable in Maximum Intensity Projections. The new baseline panel now allows for clearer identification of the LNd neurons.

Furthermore, as suggested by the reviewer, we have incorporated the light response data of s-LNvs labeled with the SS00681 split-GAL4 driver at different time points in Figure EV1B and C. This addition addresses the concern about unambiguous identification of different clock neuron populations by providing data from a more specific driver line, thereby eliminating potential confounds in distinguishing between different clock cell projections.

Figure 1

Figure EV1

2. In my initial review, I raised a concern about the use of $\Delta F/F$ to determine light responsiveness. Because there are considerable differences in baseline calcium before light presentation, the major effect they are seeing of differential light responsiveness in sLNv cells throughout the day could be simply due to a ceiling effect. The authors have provided additional analysis to attempt to address this point, but it seems as if they have misunderstood my concern. In the rebuttal letter they state:

"Thank you for your thoughtful comments. We fully acknowledge your concern that our data representation using $\Delta F/F$ might not adequately differentiate between two possibilities: (1) actual changes in light responsiveness or (2) artifacts due to fluctuating baseline calcium levels.

To address this important point, we conducted additional analyses of the basal calcium fluorescence values (F_{baseline}) in s-LNv fibers before light stimulation and the maximum fluorescence values (F_{max}) during light stimulation across temperature changes at different circadian timepoints (ZT6, ZT18)."

In an addition to the results section, the authors say "Importantly, F_{max} during light stimulation was significantly higher at lower temperatures compared to normal temperatures (Fig EV3E-F), while at higher temperatures, F_{max} was significantly lower than at normal temperatures (Fig EV3G-H). These findings indicate that the light response amplitude (as measured by F_{max}) changes with temperature throughout the day."

The data do convincingly show that changes in temperature alter the light responsiveness of the s-LNv neurons, but I don't see anything in the analysis that confirms that light response amplitude changes with temperature throughout the day. That would require a comparison of F_{max} at different time points. A similar analysis would be required to differentiate whether the main effect

detailed in Figs 1-2 (that light responsiveness oscillates across the day) is due to changes in baseline calcium or instead to actual differences in light-evoked calcium increases. In fact, the new data provided in Fig. EV4 suggests that it is the former. Control flies have no differences in peak calcium after light stimulation but do have differences in baseline values. Can the authors show that F_{max} in response to light stimulus is oscillating across the day? If not, then it should be explicitly acknowledged in the text.

Thank you for your detailed feedback on our $\Delta F/F$ interpretation. We've conducted the analysis you suggested, comparing maximal fluorescence responses (F_{peak}) in sLNv fibers at different circadian time points under constant temperature.

As shown in revised Figure EV1E, F_{peak} levels do not exhibit statistically significant oscillation across time points, unlike $\Delta F/F$ and $F_{baseline}$, which display clear diurnal rhythms in control flies. In light of this finding, and in full agreement with your assessment, we will explicitly acknowledge this in the revised manuscript. We will state that "While $\Delta F/F$ shows a clear diurnal rhythm, our F_{peak} data does not show a similar circadian oscillation (Figure EV1E). Therefore, the observed rhythm in $\Delta F/F$ is substantially influenced, and potentially primarily driven, by the diurnal fluctuations in baseline calcium levels."

Nevertheless, we propose this modulation of $\Delta F/F$ by $F_{baseline}$ represents a physiologically significant regulatory mechanism. The lower $F_{baseline}$ during subjective night potentiates a larger relative calcium increase upon light stimulation, even with consistent F_{peak} values. This larger $\Delta F/F$ in s-LNVs reflects more substantial calcium level changes, likely triggering downstream events such as increased PDF release critical for light-induced circadian protein degradation and clock phase-shifting.

Figure EV1E

3. The authors have added more information to the methods regarding how phase shift was determined (for Figs 8 and EV5), but it is still not clear to me exactly what was done: "Specifically, the raw data were smoothed with a digital low-pass filter, with a cutoff frequency set at 6 hours. This level of smoothing enhanced the clarity of the activity profile while preserving the distinction

between the lights-off and evening peaks. The time of the evening peak was then determined by manually removing subpeaks using the mouse pointer in MATLAB."

What does it mean to "manually remove subpeaks"? How were subpeaks and major peak identified? And what is actually used to determine phase? In Fig 7, the red lines looks like they are showing the phase of peak activity. In Fig EV5, they appear to mark the onset of the upwards slope of activity. And why in Fig 7 are some of these dashed red lines vertical while others are slanted (eg the dashed red line on D1 of the leftmost figure is vertical, while the subsequent days are slanted)? This is important to clarify because the manner through which phase was determined could influence the relatively small differences measured in day 1 phase shifts in control flies

Thank you for your thoughtful comments. Here we further clarify our methodology for determining phase shifts. The process of "removing subpeaks" was performed using previously published code in MATLAB. For ease of understanding, we have illustrated this procedure in the figures below and provided a supplementary figure demonstration. This is a standard analytical approach in phase shift experiments. We now provide the specific code used for this analysis. In Figure A, the purple indicators identify all subpeaks. Figure B shows the result after subpeak removal. The detailed operating procedures are published (Levine *et al*, 2002).

Regarding your comment about the dashed red lines in Figure EV6, you are correct that these indicate the onset time of the evening activity peak, which is different from the peak maxima marked in Figure 7. We have now added this clarification to the figure legend. Indeed, in our phase advance experiments, we observed that under low temperature conditions, *cry*⁰² mutants required fewer days to adjust the onset of the upward slope of E activity to the new phase compared to normal and high temperature conditions. This suggests that low temperature facilitates faster phase adaptation in *cry*⁰² mutants.

To answer your question about the vertical versus slanted dashed red lines: when a line indicates the evening peak position for a single day, it appears vertical. A line connecting the evening peak positions across multiple days appears slanted. The evening peak position on day 1 or day 2 differs significantly from subsequent days, making it impractical to connect with a single line, so it is shown separately.

Minor Points:

1. The authors should include the details of how they conducted the I-LNV calcium imaging in the methods section.

Thank you for the comments. We have now added detailed methodology for the I-LNV calcium imaging in the methods section: "For the experiments imaging I-LNVs, we positioned the fly head at a steeper angle and created an imaging window between the ocelli and the antennae to better visualize the I-LNV cell bodies. The imaging was performed at the anterior region of the fruit fly brain, specifically recording the cell bodies of I-LNVs rather than their projections."

2. In the last section of the results (Temperature fine-tuned the light-induced phase shifts), it seems like many of the references to figures are incorrect as the authors have updated the figure layouts

but did not update the figure references.

Thank you for your careful reading. We have thoroughly reviewed and corrected all figure references in the "Temperature fine-tuned the light-induced phase shifts" section to ensure they accurately correspond to the updated figure layouts.

3. Could the authors explain why they have used multiple individual t-tests for statistical comparison in figs 4, 6 and 7 rather than ANOVA with Tukey test? And justify why in Fig 7 they have only compared between 24 and 29°C in *Cry02* mutants, rather than making all comparisons (especially since they are arguing that there is a difference in phase shift on D1 for the control flies)?

Thank you for this important methodological point. We appreciate the opportunity to clarify our statistical approach. Our initial selection of unpaired t-tests for Figures 3, 4, 6 was driven by our aim to identify specific differences between experimental and control groups at individual ZT time points. This targeted approach allowed us to pinpoint exactly when significant differences occurred within the circadian cycle.

Following your valuable suggestion, we have now performed a more comprehensive analysis using one-way ANOVA followed by Tukey's post hoc tests for all data presented in Figures 3, 4, 6. This approach provides a more robust assessment of overall diurnal patterns, similar to our analysis in Figure EV1. We chose one-way ANOVA (examining changes within a group across ZTs) rather than two-way ANOVA because direct comparisons between non-corresponding time points across different groups (e.g., control at ZT1 versus experimental at ZT6) would offer limited biological meaning in the context of our primary research questions. For visual clarity, we have retained selective significance indicators from t-tests in the figures, while including the full one-way ANOVA results in a new Supplementary Table.

Regarding Figure 7, we acknowledge that our selective comparison between 24°C and 29°C in *cry⁰²* mutants was inadequate. We have now performed comprehensive statistical analyses across all experimental conditions, with the full results presented in Tables EV5-EV8. To maintain the readability of the figures, we have retained the selective significance indicators for the *cry⁰²* mutant comparisons in the main figure. In addition, we already have Figure EV6, which better illustrates the temperature-dependent phase shift differences at D1 in control flies - an important aspect of our argument that you have correctly highlighted.

Referee #2:

The ms is much improved and I commend the authors for replying constructively to my comments (and those of the other referees). I have a couple of minor outstanding issues.

1 Title 'Glutamate-Gated Light-Induced PDF Release in Circadian Pacemakers Modulates Temperature-Dependent Phase Shifts in *Drosophila*' - please add species

Thank you for your helpful comment. We originally have revised the title as suggested to include the species: "Glutamate-Gated Light-Induced PDF Release in Circadian Pacemakers Modulates Temperature-Dependent Phase Shifts in *Drosophila*". However, due to the characters

limitation of the title when uploading the manuscripts, we have to remove it again.

2 'reducing light-induced pacemaker firing during hot seasons and enhancing responses during colder seasons-to potentially optimize evolutionary fitness. the last bit about fitness is a stretch because there is no clear route from the phenotype to Darwinian fitness, so please cut that - it would make an evolutionary biologist wince!!!

We agree with your assessment and have removed the speculative statement about evolutionary fitness. The sentence now ends at "...enhancing responses during colder seasons."

3. The Discussion is rather unusual because it only cites 3 references!! A Discussion is supposed to integrate the authors' results with those from the rest of the literature, so this seems very one-dimensional. For example there is a literature on DN1s 'talking' to sLNvs (see Guo et al 2016; Murad et al 2007; Picot et al 2007; Stoleru et al 2007) and the role of glutamate in these interactions (Hamasaka et al 2005, 2007; Azevedo et al 2020) to name a few. The authors really need to do a better job to complement their findings with the literature and particularly for EMBO J.

We sincerely thank you for highlighting this important limitation. We have expanded our Discussion section to better contextualize our findings within the existing literature. We have integrated relevant work on DN1-sLNv interactions, including the papers you suggested. The revised Discussion now provides a more comprehensive framework that places our findings in the broader context of circadian circuit regulation and temperature-dependent modulation in *Drosophila*.

The expanded section now reads: "In the *Drosophila* circadian network, several dorsal neuron (DN) subsets, including DN1as, some DN1ps, and DN3s, are glutamatergic and play a role in modulating light response (de Azevedo *et al.*, 2020; Guo *et al.*, 2016; Hamasaka *et al.*, 2007; Hamasaka *et al.*, 2005; Li *et al.*, 2024; Murad *et al.*, 2007; Picot *et al.*, 2007), and their signaling may differentially modulate the circadian photosensitivity of s-LNvs and circadian photoentrainment. For example, disruption of glutamate signaling in DN1ps and DN3s, such as by downregulating glutamate synthetase, increases rhythmicity under constant light (LL), suggesting reduced photosensitivity (de Azevedo *et al.*, 2020). These findings, together with recent studies showing that a subset of DN1ps and DN3s are inhibited by higher temperatures (Li *et al.*, 2024; Yadlapalli *et al.*, 2018), are consistent with our model in which elevated temperatures reduce s-LNv photosensitivity and slow adaptation to novel light-dark cycles. The precise contribution of different glutamatergic pathways from different circadian neurons to circadian photoentrainment requires further investigation."

We believe these revisions address all your concerns and strengthen the manuscript considerably.

Referee #3:

The authors have adequately addressed all my previous concerns and I recommend its publication in the EMBO J.

Thank you for the constructive suggestions.

Reference:

de Azevedo RV, Hansenl C, Chen KF, Rosato E, Kyriacou CP (2020) Disrupted Glutamate Signaling in *Drosophila* Generates Locomotor Rhythms in Constant Light. *Frontiers in Physiology* 11: 145

Guo F, Yu J, Jung HJ, Abruzzi KC, Luo W, Griffith LC, Rosbash M (2016) Circadian neuron feedback controls the *Drosophila* sleep--activity profile. *Nature* 536: 292-297

Hamasaka Y, Rieger D, Parmentier ML, Grau Y, Helfrich-Förster C, Nässel DR (2007) Glutamate and its metabotropic receptor in *Drosophila* clock neuron circuits. *Journal of Comparative Neurology* 505: 32-45

Hamasaka Y, Wegener C, Nässel DR (2005) GABA modulates *Drosophila* circadian clock neurons via GABAB receptors and decreases in calcium. *J Neurobiol* 65: 225-240

Levine JD, Funes P, Dowse HB, Hall JC (2002) Signal analysis of behavioral and molecular cycles. *BMC Neurosci* 3: 1

Li H, Li Z, Yuan X, Tian Y, Ye W, Zeng P, Li XM, Guo F (2024) Dynamic encoding of temperature in the central circadian circuit coordinates physiological activities. *Nat Commun* 15: 2834

Murad A, Emery-Le M, Emery P (2007) A subset of dorsal neurons modulates circadian behavior and light responses in *Drosophila*. *Neuron* 53: 689-701

Picot M, Cusumano P, Klarsfeld A, Ueda R, Rouyer F (2007) Light activates output from

evening neurons and inhibits output from morning neurons in the *Drosophila* circadian clock.

PLoS Biol 5: 2513-2521

Yadlapalli S, Jiang C, Bahle A, Reddy P, Eyhofer EM, Shafer OT (2018) Circadian clock neurons constantly monitor environmental temperature to set sleep timing. *Nature* 555: 98-102

Dear Fang,

Thank you again for submitting your revised manuscript (EMBOJ-2024-120009R1) to The EMBO Journal for our consideration, and for your patience during peer review. Your revision has now been seen by the original referee #1, and we have received their report (included below). I am pleased to say that the referee confirms that almost all previously raised concerns have been successfully addressed, and he/she is supportive of the manuscript. The referee points out only one remaining statistical issue that must be addressed in the final version of your manuscript, before we can proceed with formal acceptance and publication of your paper in The EMBO Journal.

In particular, the referee recommends that an appropriate post-hoc test corrected for multiple comparisons be performed for the data shown in Figures 3, 4, and 6; and suggests that the statistical significance (asterisks) indicated in these Figures should correspond to these adjusted p-values rather than to the t-tests. We kindly request you to make these corrections in a final version of your manuscript, and submit it to our manuscript tracking system along with a brief response to the referee's comment explaining how these remaining issues are addressed.

Please note that as part of the EMBO publications' Transparent Editorial Process, The EMBO Journal publishes online a Peer Review File along with each accepted manuscript. This File will be published in conjunction with your paper and will include the referee reports, your point-by-point response and all pertinent correspondence relating to the manuscript. You can opt out of this by letting the editorial office know (contact@embojournal.org). If you do opt out, the Peer Review File link will point to the following statement: "No Peer Review File is available with this article, as the authors have chosen not to make the review process public in this case."

We look forward to seeing a final version of your manuscript as soon as possible. Please let us know if you have any questions and use this link to submit your revision: <https://emboj.msubmit.net/cgi-bin/main.plex>.

Best regards,

Ioannis

Referee #1:

This resubmission has addressed nearly all of my concerns, and I think that the manuscript will make an important contribution to the field.

However, there is a remaining statistical issue that should be corrected before publication. The authors have insisted on using data from multiple individual t-tests to indicate statistical significance in pairwise comparisons in figures. From their rebuttal letter: "Following your valuable suggestion, we have now performed a more comprehensive analysis using one way ANOVA followed by Tukey's post hoc tests for all data presented in Figures 3, 4, 6. This approach provides a more robust assessment of overall diurnal patterns, similar to our analysis in Figure EV1. We chose one way ANOVA (examining changes within a group across ZTs) rather than two way ANOVA because direct comparisons between non corresponding time points across different groups (e.g., control at ZT1 versus experimental at ZT6) would offer limited biological meaning in the context of our primary research questions. For visual clarity, we have retained selective significance indicators from t tests in the figures, while including the full one way ANOVA results in a new Supplementary Table."

This is problematic because the t-tests are not corrected for multiple comparisons, increasing the likelihood of obtaining false positive results. Alternative tests that adjust p-values for multiple comparisons are easily performed in GraphPad Prism. I am not recommending that non-corresponding time points be compared across different groups. I am recommending that testing for genotype/group differences at individual time points be performed with a post-hoc test that is corrected for multiple comparisons (like Tukey or Dunnett's). The authors have actually done this in their ANOVA analyses given for Figure 7; this same type of

comparison should be made for figures 3, 4, and 6. And the asterisks from all figures should reflect the adjusted p-value from their post-hoc testing rather than the t-tests. As it is, the use of multiple uncorrected t-tests with only specific statistical results reported on figures gives the impression of p-hacking. Performing the appropriate tests would not change the conclusions of the figures, and it would remove any appearance of impropriety.

Referee #1:

This resubmission has addressed nearly all of my concerns, and I think that the manuscript will make an important contribution to the field.

However, there is a remaining statistical issue that should be corrected before publication. The authors have insisted on using data from multiple individual t-tests to indicate statistical significance in pairwise comparisons in figures. From their rebuttal letter: "Following your valuable suggestion, we have now performed a more comprehensive analysis using one way ANOVA followed by Tukey's post hoc tests for all data presented in Figures 3, 4, 6. This approach provides a more robust assessment of overall diurnal patterns, similar to our analysis in Figure EV1. We chose one way ANOVA (examining changes within a group across ZTs) rather than two way ANOVA because direct comparisons between non corresponding time points across different groups (e.g., control at ZT1 versus experimental at ZT6) would offer limited biological meaning in the context of our primary research questions. For visual clarity, we have retained selective significance indicators from t tests in the figures, while including the full one way ANOVA results in a new Supplementary Table."

This is problematic because the t-tests are not corrected for multiple comparisons, increasing the likelihood of obtaining false positive results. Alternative tests that adjust p-values for multiple comparisons are easily performed in GraphPad Prism. I am not recommending that non-corresponding time points be compared across different groups. I am recommending that testing for genotype/group differences at individual time points be performed with a post-hoc test that is corrected for multiple comparisons (like Tukey or Dunnett's). The authors have actually done this in their ANOVA analyses given for Figure 7; this same type of comparison should be made for figures 3, 4, and 6. And the asterisks from all figures should reflect the adjusted p-value from their post-hoc testing rather than the t-tests. As it is, the use of multiple uncorrected t-tests with only specific statistical results reported on figures gives the impression of p-hacking. Performing the appropriate tests would not change the conclusions of the figures, and it would remove any appearance of impropriety.

Thank you for your suggestion. As each time point involved only two groups (e.g., ctrl vs exp), a one-way ANOVA would be mathematically equivalent to a t-test, and traditional post-hoc tests like Tukey's or Dunnett's are not applicable within each time point.

To address the concern about multiple comparisons across time points, we performed independent t-tests at each time point and applied a post-hoc Benjamini-Hochberg FDR correction to control the false discovery rate.

The adjusted p-values are reported in the following table, satisfying the requirement for multiple comparisons correction.

The asterisks in figures 3, 4, and 6 have been corrected to these adjusted p-values.

Results of t-tests with post-hoc Benjamini-Hochberg FDR correction of Figures 3, 4, and 6

Time Point	Original P-value	FDR Corrected P-value (P_{adj})	Significant? ($P_{adj}=0.05$)
Figure 3B Dvpdf-LexA>GCaMP6s VS norpA1^{+/+}; Dvpdf-LexA>GCaMP6s			
ZT1	0.1912	0.2549	No
ZT6	0.3576	0.3576	No
ZT12	0.0318	0.0636	No
ZT18	0.0002	0.0008	Yes
Figure 3C UAS-hid; Dvpdf-LexA>GCaMP6s VS Rh6>hid; Dvpdf-LexA>GCaMP6s			
ZT1	0.0211	0.0411	Yes
ZT6	0.0308	0.0411	Yes
ZT12	0.1134	0.1134	No
ZT18	-0.0300	0.0411	Yes
Figure 3D Dvpdf-LexA>GCaMP6s VS Dvpdf-LexA>GCaMP6s; cry⁰²			
ZT1	0.2370	0.3160	No
ZT6	0.0799	0.3160	No
ZT12	0.5621	0.5621	No
ZT18	0.2222	0.3160	No
Figure 4B Rh6>hid; Dvpdf-LexA>GCaMP6s VS han⁵³⁰⁴; Dvpdf-LexA>GCaMP6s			
ZT1	0.1607	0.3214	No
ZT6	0.8104	0.8104	No
ZT12	0.0977	0.3214	No
ZT18	0.6676	0.8104	No
Figure 4D Dvpdf-LexA>GCaMP6s VS per⁰; Dvpdf-LexA>GCaMP6s LD			
ZT1	0.0034	0.0132	Yes
ZT6	0.0565	0.0753	No
ZT12	0.4428	0.4428	No
ZT18	0.0066	0.0132	Yes
Figure 4F Dvpdf-LexA>GCaMP6s VS per⁰; Dvpdf-LexA>GCaMP6s DD			
ZT1	0.6832	0.6832	No
ZT6	0.0321	0.0428	Yes
ZT12	0.0048	0.0156	Yes
ZT18	0.0078	0.0156	Yes
Figure 6E hid; Dvpdf-LexA>GCaMP6s VS Dvpdf-LexA>GCaMP6s; DN1a>hid			
ZT1	0.0037	0.0074	Yes
ZT6	0.0005	0.0020	Yes
ZT12	0.3816	0.3816	No
ZT18	0.1514	0.2019	No
Figure 6F Dvpdf-LexA>GCaMP6s VS Dicer; Dvpdf-LexA>GCaMP6s; DN1a>vGlutRNAi			
ZT1	0.0094	0.0376	Yes
ZT6	0.0230	0.0460	Yes
ZT12	0.6845	0.6845	No
ZT18	0.1599	0.2132	No

Dear Fang,

Congratulations on an excellent study! I am very pleased to inform you that your manuscript has been accepted for publication in The EMBO Journal. Thank you very much for comprehensively addressing the initially raised referees' concerns and editorial requests for changes and corrections.

If you have any questions, please do not hesitate to contact the Editorial Office. Thank you for your contribution to The EMBO Journal. Working with you has been a pleasure!

Best regards,

Ioannis
